# Exploring the most promising anti - Depressant drug targeting Microtubule Affinity Receptor Kinase 4 involved in Alzheimer's Disease through molecular docking and molecular dynamics simulation

S. Rehan Ahmad[1]*, Md. Zeyaullah[2]*, Abdullah M. AlShahrani[2], Mohammad Suhail Khan[3], Khursheed Muzammil[3], Faheem Ahmed[3], Adam Dawria[3], Ali Mohieldin[3], Haroon Ali[3], Abdelrhman A. G. Altijani[3]

1 Hiralal Mazumdar Memorial College for Women, West Bengal State University, Kolkata, West Bengal, India, 2 Department of Basic Medical Science, College of Applied Medical Sciences, Khamis Mushait Campus, King Khalid University (KKU), Abha, Saudi Arabia, 3 Department of Public Health, College of Applied Medical Sciences, Khamis Mushait Campus, King Khalid University (KKU), Abha, Saudi Arabia

* mdhafed@kku.edu.sa (MZ); professor.rehaan@gmail.com (SRA)

**Data Availability Statement:** All relevant data are within the manuscript.

## Abstract

Alzheimer's Disease (AD) is the prevailing type of neurodegenerative illness, characterised by the accumulation of amyloid beta plaques. The symptoms associated with AD are memory loss, emotional variability, and a decline in cognitive functioning. To date, the pharmaceuticals currently accessible in the marketplace are limited to symptom management. According to several research, antidepressants have demonstrated potential efficacy in the management of AD. In this particular investigation, a total of 24 anti-depressant medications were selected as ligands, while the Microtubule Affinity Receptor Kinase 4 (MARK4) protein was chosen as the focal point of our study. The selection of MARK4 was based on its known involvement in the advancement of AD and other types of malignancies, rendering it a highly prospective target for therapeutic interventions. The initial step involved doing ADMET analysis, which was subsequently followed by molecular docking of 24 drugs. This was succeeded by molecular dynamics simulation and molecular mechanics generalised Born surface area (MMGBSA) calculations. Upon conducting molecular docking experiments, it has been determined that the binding affinities observed fall within the range of -5.5 kcal/mol to -9.0 kcal/mol. In this study, we selected six anti-depressant compounds (CID ID - 4184, 2771, 4205, 5533, 4543, and 2160) based on their binding affinities, which were determined to be -9.0, -8.7, -8.4, -8.3, -8.2, and -8.2, respectively. Molecular dynamics simulations were conducted for all six drugs, with donepezil serving as the control drug. Various analyses were performed, including basic analysis and post-trajectory analysis such as free energy landscape (FEL), polarizable continuum model (PCM), and MMGBSA calculations. Based on the findings from molecular dynamics simulations and the MMGBSA analysis, it can be inferred that citalopram and mirtazapine exhibit considerable potential as anti-depressant

**Funding:** Deanship of Scientific Research at King Khalid University, Kingdom of Saudi Arabia, Research Project Grant Number RGP2/355/44.

**Competing interests:** The authors have declared that no competing interests exist.

**Abbreviations:** AD, Alzheimer's Disease; SSRI, selective serotonin reuptake inhibitors; MARK4, microtubule affinity regulation kinase 4; RMSD, root mean square deviation; RMSF, root mean square fluctuation; SASA, solvent Accessible Surface Area; FEL, free Energy Landscape; PCA, Principal Component investigation; MMGBSA, molecular mmechanics generalised Born ssurface area.

agents. Consequently, these compounds warrant further investigation through *in vitro* and *in vivo* investigations in the context of treating AD.

## Introduction

Alzheimer's disease (AD) is a rapidly expanding neurodegenerative condition characterised by progressive and irreversible deterioration of the central nervous system, resulting in cognitive and behavioural deficits [1–3]. The process of diagnosing a condition relies on the analysis of historical case data, clinical evaluations, neuroimaging techniques, and blood analyses. The brain has two primary categories of aberrant proteins, namely amyloid-beta (Aβ) and tau. The disease advances as a consequence of the accumulation of these proteins [4–7]. At now, the therapeutic options for AD are limited to a total of five drugs. Among them, four belong to the class of cholinesterase inhibitors, while the last prescription, memantine, functions by inhibiting glutamate formation. Cholinesterase inhibitors are administered to patients diagnosed with mild to moderate AD, while memantine is prescribed for individuals in the mild to severe stages of AD [8–10].

In the present work, the researchers examined the possibility of microtubule affinity regulation kinase 4 (MARK4) as a target for AD. Previous studies have indicated an observed overexpression of MARK 4 in AD, and this overexpression has been linked to the progression of AD. The discovery of the crystal structure of MARK4 was facilitated by the utilisation of the pyrazolopyrimidine inhibitor [11]. The overexpression of MARK4 has been found to be accountable for the phosphorylation of the tau protein at the Ser262 location, a crucial step in the binding of microtubules to tau proteins [11]. In their study, Lund et al. 2014 observed the co-localization of phosphorylated MARK4 and phosphor-tau Ser262 in granulomatous (GVD) formations found in samples of AD [12]. The findings of this study proposed the development of a very effective MARK4 inhibitor for the treatment of AD [12]. In the context of this study, it is posited that MARK4 represents the most suitable candidate for addressing neurodegeneration in AD.

AD is widely regarded as a condition that lacks a definitive cure. To date, a total of 24 selective serotonin reuptake inhibitors (SSRIs) have been examined in the existing literature as potential treatments for some symptoms associated with major depressive disorder [13]. The development of AD is frequently associated with the early manifestation of depression, characterised by a chronic relapsing-remitting pattern that tends to worsen and become more extended as individuals grow older. This progression significantly heightens the probability of acquiring AD. Late-life depression has been found to be associated with reduced levels of neurotrophins, particularly brain-derived neurotrophic factor. Additionally, it is characterised by the activation of neuroinflammatory pathways, heightened secretion of pro-inflammatory cytokines and C-reactive protein, and an increased cortical amyloid burden. These factors collectively contribute to the process of neurodegeneration. Historically, there was a prevailing belief that the sole determinant of the symptoms associated with AD was cholinergic dysfunction, as evidenced by references [14–16]. Geeldenhuys, Van der Schyf, and Ramirez have highlighted the significance of the 5-Hydroxytryptamine (5-HT)1, 5-HT4, 5-HT6, and 5-HT7 receptor classes in the context of cognitive enhancement. In comparison to conventional antidepressants such as tricyclics, selective serotonin reuptake inhibitors (SSRIs) exhibit a more desirable side effect profile and have been authorised as primary therapeutic interventions for depressive disorders [14–16]. There exists compelling data indicating that SSRIs have

demonstrated a beneficial impact in delaying the onset of AD in those with a prior history of depression. In addition to the reduction of serotonin transporter function, newly developed serotonergic antidepressants such as vortioxetine exert an impact on serotonin receptors, specifically through the antagonism of 5-HT7 receptors. Vortioxetine significantly improved cognition when compared to other traditional anti-depressants in depressed adults with moderate AD when compared to placebo [17]. Based on the recent meta-analysis finding, sertraline and mirtazapine can be considered as an alternative treatment for depression in AD [18]. Even the combination of antipsychotics (such as risperidone and quetiapine) and mirtazapine can be used for the management of AD [19, 20].

Upon reviewing the previous successful results of SSRIs in managing AD symptoms, the present study conducted molecular docking analyses for a total of 24 antidepressant compounds. Subsequently, utilising the top five binding affinities, molecular dynamics simulations were performed for all six SSRI drugs (CID ID - 4184, 2771, 4205, 5533, 4543, and 2160). Both CID 2160 and CID 4543 were subjected to simulation as their binding affinities were found to be same. CID ID-3152 (Donepezil), an authorised medicine for AD, was selected as the control in our study. The primary objective of this study is to identify the most efficacious SSRI medication that specifically targets the MARK4 protein implicated in AD.

**Table 1. Name of 24 anti-depressant drugs collected from literature.**

| Drug Name | Mode of action | CID ID | Reference |
|---|---|---|---|
| Sertraline | SRI | 68617 | 13 |
| Fluoxetine | SRI | 3386 | 13 |
| Escitalopram | SRI | 146570 | 13 |
| Fluvoxamine | SRI | 5324346 | 13 |
| Paroxetine | SRI | 43815 | 13 |
| Citalopram | SRI | 2771 | 13 |
| Agomelatine | M1,2 + 5-HT2c | 82148 | 13 |
| Mianserin | 5-HT2, a1, a2 | 4184 | 13 |
| Reboxetine | NRI | 127151 | 13 |
| Trazodone | 5-HT2, a1, SRI | 5533 | 13 |
| Venlafaxine | SRI+NRI | 5656 | 13 |
| Bupropion | NRI+DRI | 444 | 13 |
| Duloxetine | SRI+NRI | 60835 | 13 |
| Mirtazapine | 5-HT2, 5-HT3, a2 | 4205 | 13 |
| Tranylcypromine | Irreversible MAOI | 5530 | 13 |
| Moclobemide | Reversible MAOI | 4235 | 13 |
| Phenelzine | Irreversible MAOI | 3675 | 13 |
| Isocarboxazid | Irreversible MAOI | 3759 | 13 |
| Amitriptyline | SRI+NRI | 2160 | 13 |
| Trimipramine | SRI+NRI | 5584 | 13 |
| Imipramine | SRI+NRI | 3696 | 13 |
| Clomipramine | SRI | 2801 | 13 |
| Dosulepin | SRI+NRI | 5284550 | 13 |
| Nortriptyline | NRI | 4543 | 13 |

## Methodology

### 1. Selection of SSRI drugs

We took 24 anti-depressant drugs from the literature based on their mode of action as well as easy availability and well establishment in the global market and it is shown in Table 1.

### 2. ADMET analysis of 24 SSRI drugs

The pharmacokinetics and Lipinski's rule are central to accelerating drug discovery and development. As 24 SSRI drugs are already in the market, but still we performed ADMET analysis with the SWISS ADME server which is shown in S1 Table and we even plotted the radar graph for the the binding affinity of the top six drugs and Donepezil.

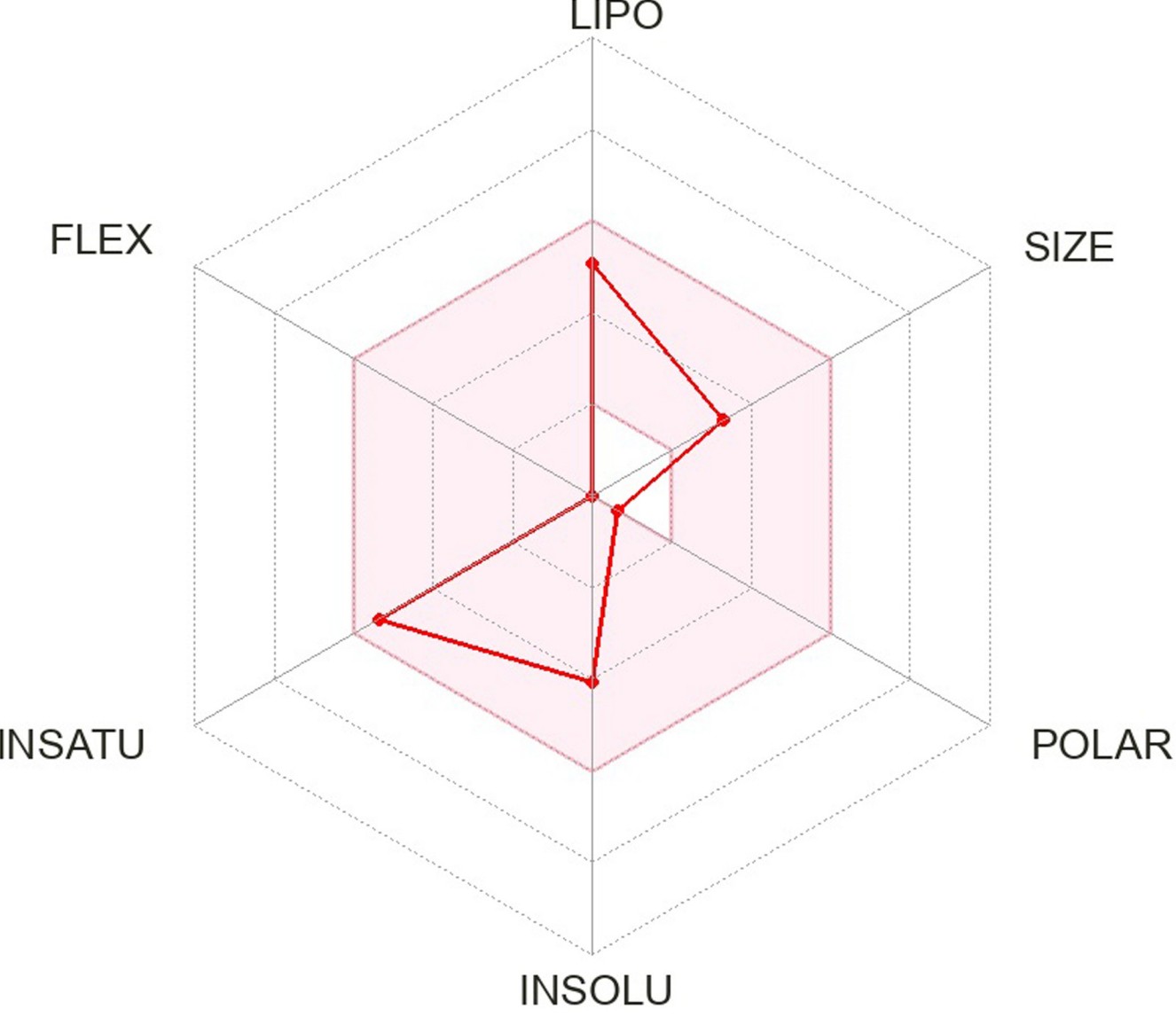

**Fig 1. Radar plots of six SSRI drugs - (1) 4184 (2) 2771 (3) 4205 (4) 5533 (5) 4543 (6) 2160 (7)3152.**

### 3. Molecular docking of 24 SSRI drugs

Molecular docking was performed with AutoDock 4.2.6. First, we converted PDB structures to PDBQT format, added Gasteiger charges to the ligands, and then loaded MARK4-associated protein (5ES1) to AutoDock MGL tools. A grid box with x, y, and z directions was generated to cover the active sites of the target protein, keeping enough space for the rotational and translational movement in the ligands and keeping the grid spacing at 0.375 Å, and flexible docking was performed. The outputs were analysed and visualised in AutoDock 4.2.6.

### 4. Molecular dynamics simulation of 6 SSRI drugs

Desmond, a program from Schrödinger LLC [21], was used to assess the effect of the solvation model on the binding affinity between the best-screened ligands and target protein.

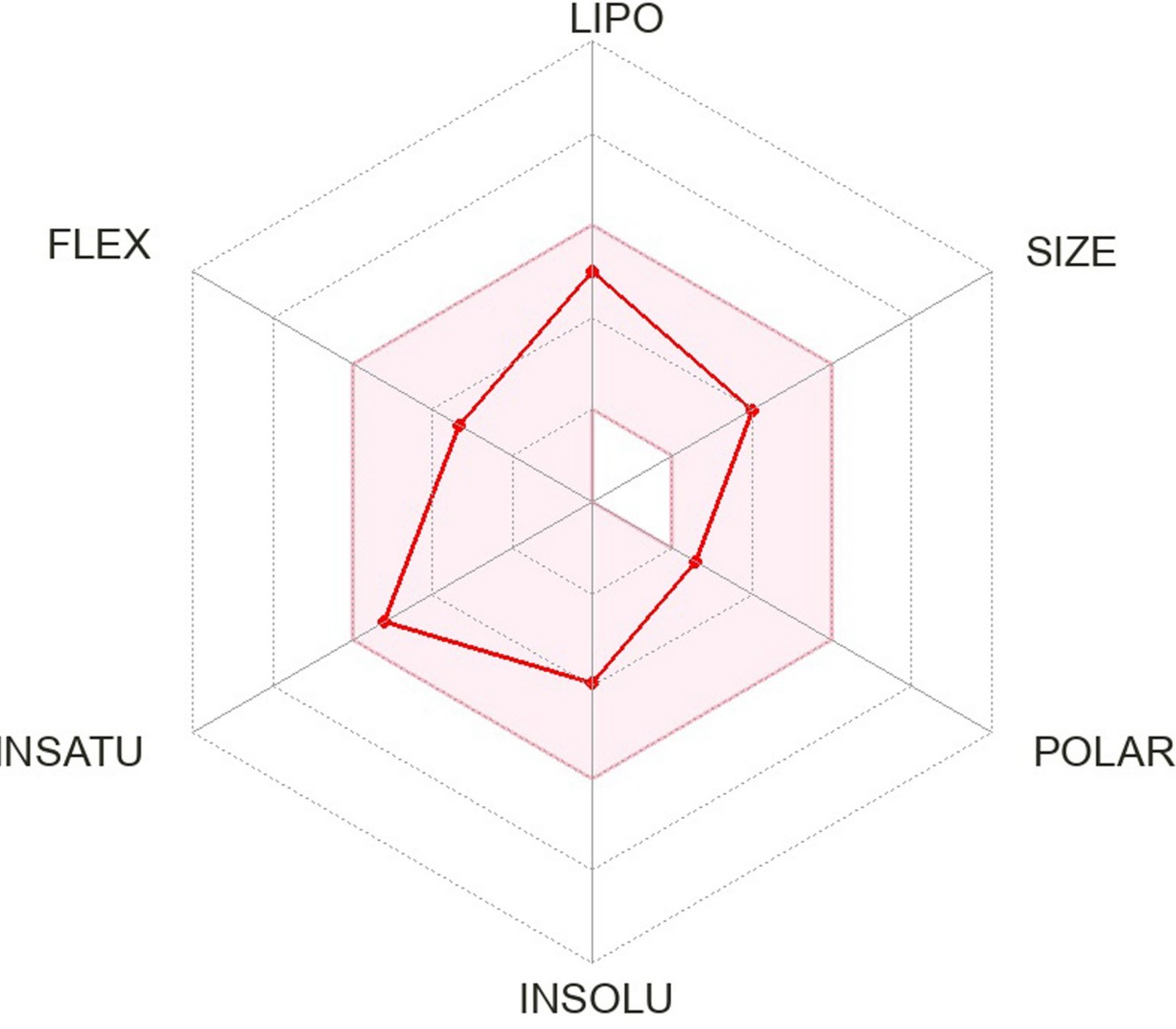

**Fig 2. Radar plots of six SSRI drugs - (1) 4184 (2) 2771 (3) 4205 (4) 5533 (5) 4543 (6) 2160 (7)3152.**

Molecular docking is a static representation of interactions between protein and ligand in a vacuum [22]. In contrast, molecular dynamics simulations use Newton's classical equation of motion to compute atom motions over time. An attempt was made to assess the binding status of all complexes with the target protein (MARK4; PDB ID:5ES1) in the physiological environment predicted using simulations [23, 24]. The target protein complexes with the drugs were subjected to molecular dynamics simulations for 100 ns each. Pre-processing of the protein-ligand complexes was done using the Protein Preparation Wizard of Maestro software to optimise and minimise the structure of the protein-ligand complexes. All of the systems were prepared using the System Builder tool. TIP3P water (Transferable Intermolecular Interaction Potential 3 Points) was chosen as a solvent model with an orthorhombic box. The simulation was run using the OPLS 2005 force field [25]. To simulate physiological conditions, 0.15 M sodium chloride (NaCl) was added to the solvated complexes to

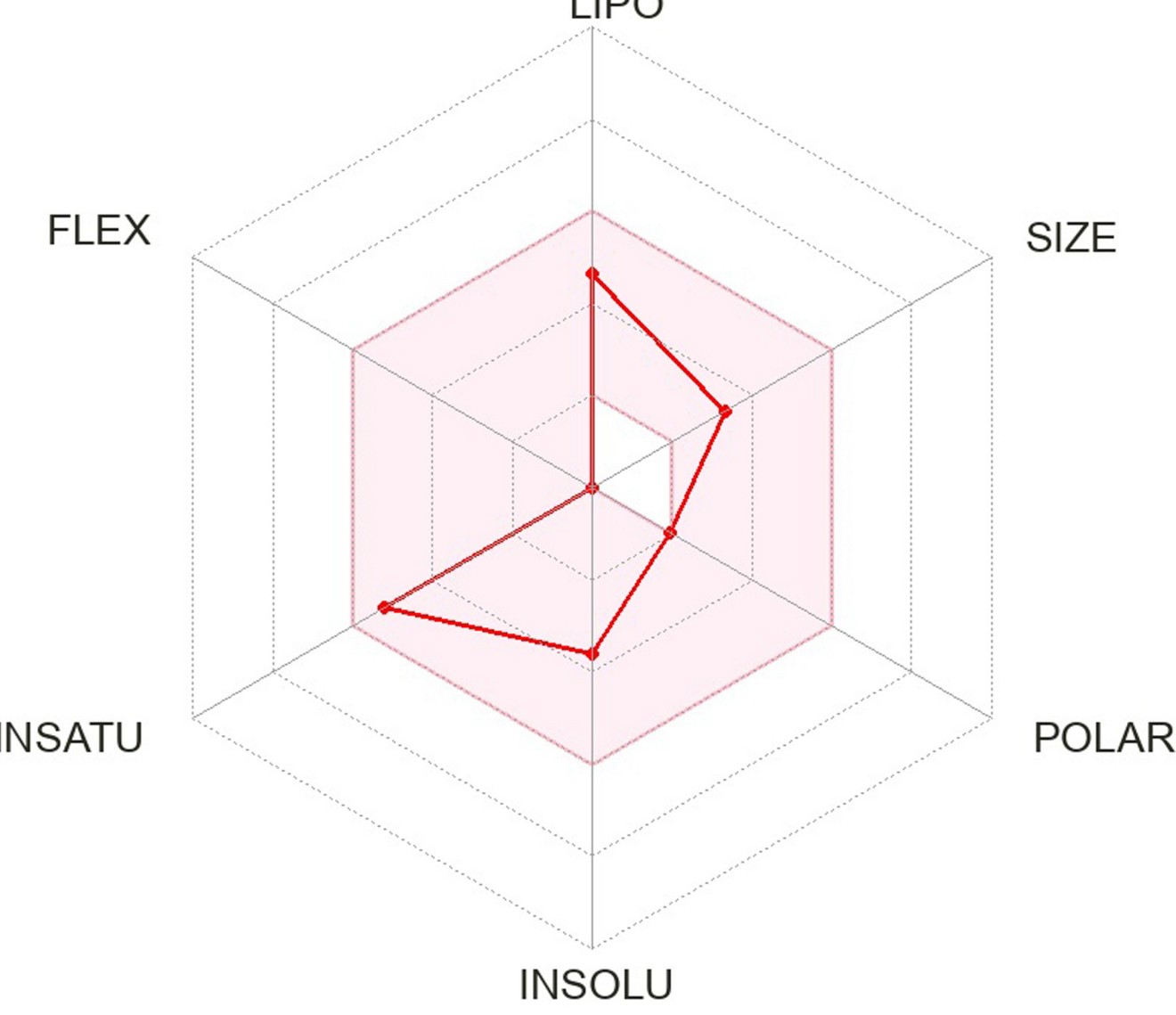

**Fig 3. Radar plots of six SSRI drugs - (1) 4184 (2) 2771 (3) 4205 (4) 5533 (5) 4543 (6) 2160 (7)3152.**

neutralise the system. The NPT ensemble was used throughout the simulation, with a temperature of 300 K and a pressure of 1 atm.

## 5. Molecular Mechanics Generalised Born Surface Area (MMGBSA) calculations

The binding free energies (Gobind) of the anchored complexes were calculated using the first Molecular Mechanics Generalized Born Surface Area (MMGBSA) module (Schrodinger suite, LLC, New York). Binding free energies were calculated using the OPLS 2005 force field, VSGB solvent model and rotamer search method [26]. After performing MD, 10

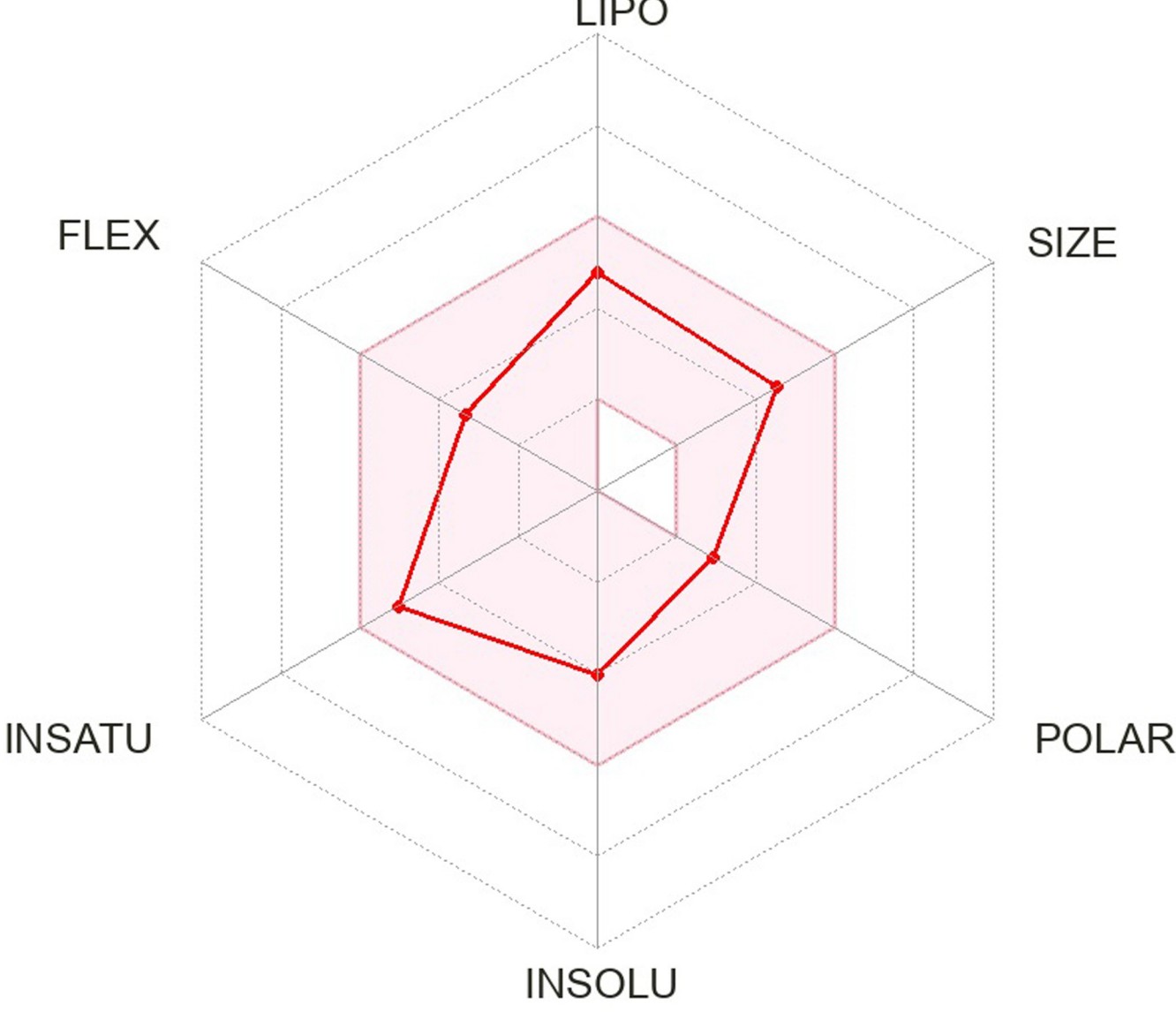

**Fig 4. Radar plots of six SSRI drugs - (1) 4184 (2) 2771 (3) 4205 (4) 5533 (5) 4543 (6) 2160 (7)3152.**

sec was used to select the MD orbital frame. The total free binding energy is calculated using the below-mentioned formula:

$$\Delta Gobind = Gcomplex - (G - protein + Gligand)$$

$\Delta Gobind$ = binding free energy, Gcomplex = free energy of the complex, G-protein = free energy of the target protein, and Gligand = free energy of the ligand.

Ethical Statements: This article does not contain any studies involving human or animal participants performed by any of the authors.

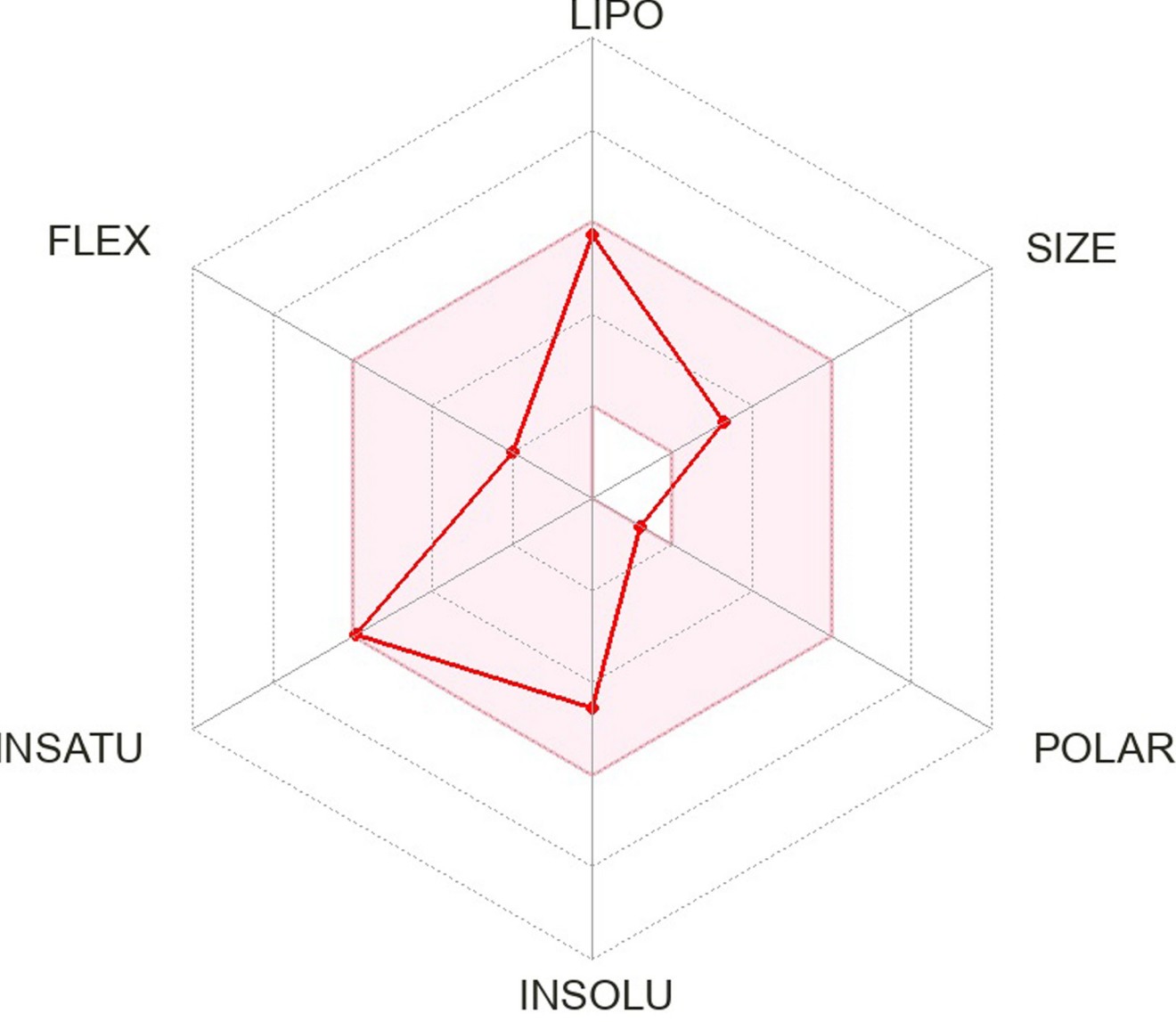

**Fig 5. Radar plots of six SSRI drugs - (1) 4184 (2) 2771 (3) 4205 (4) 5533 (5) 4543 (6) 2160 (7)3152.**

## Results

### 1. Pharmacokinetics and drug likeliness profile of 24 SSRI drugs

Lipinski's is one of the most essential drug discovery and development factors. In this study all the 24 drugs are available in the market, which means drugs follow Lipinski's rule.In this study, we generated radar plots for the binding affinity of the top six drugs (CID ID - 4184, 2771, 4205, 5533, 4543, and 2160) lies between -9 to -8.2 kcal/mol as well as radar plot of Donepezil which is already approved for AD. We use the SWISS-ADME web server for which the requirement is the SMILES notation of the selected compounds, which is used as an input based on the database predicting their pharmacokinetic property and plotted the results as a

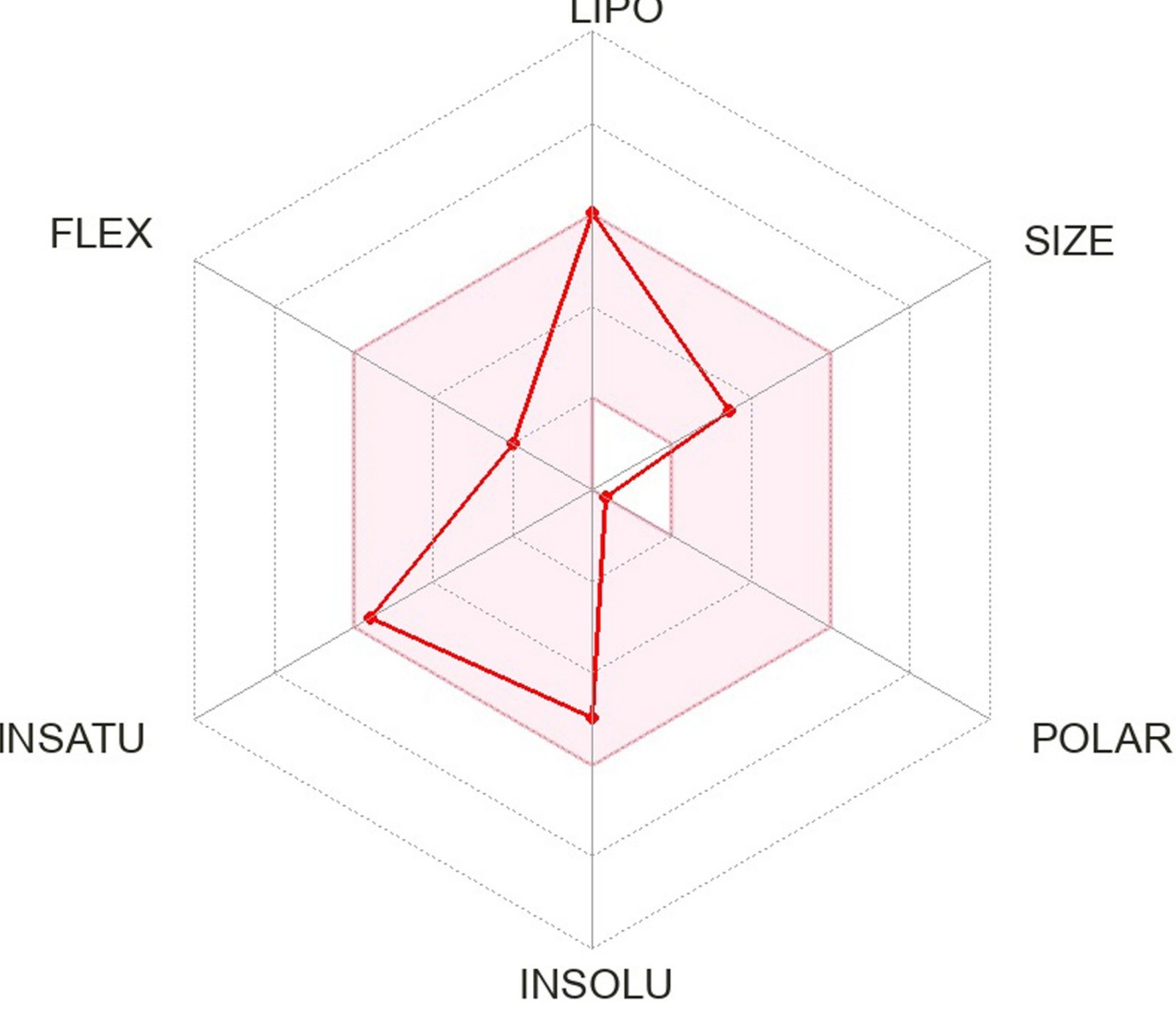

**Fig 6. Radar plots of six SSRI drugs - (1) 4184 (2) 2771 (3) 4205 (4) 5533 (5) 4543 (6) 2160 (7)3152.**

radar plot through which we can analyse the compound's lipophilic property, size, polarity, insolubility, instauration and flex properties. The more it is towards the centre, the more it is considered a good result. The radar plots are shown as Figs 1–7.

## 2. Molecular docking analysis of 24 SSRI drugs

First, molecular docking was conducted to validate the intermolecular interactions between the MARK4 protein and 24 drugs. The docking results were arranged in the order of lowest binding energy with each target protein, as shown in Table 2. The binding affinity of the top six drugs lies between -9 to -8.2 kcal/mol. After analysing the binding modes for the top six protein-ligand complexes, 5SE1-2160 formed an h-bond interaction with ASP196 with a bond length of 2.8 Å. 5SE1-2771 formed an h-bond interaction with ILE62 with a bond length of 2.8 Å. 5SE1-3152 formed an h-bond interaction with GLY65 and ILE62 with bond lengths of 4.0

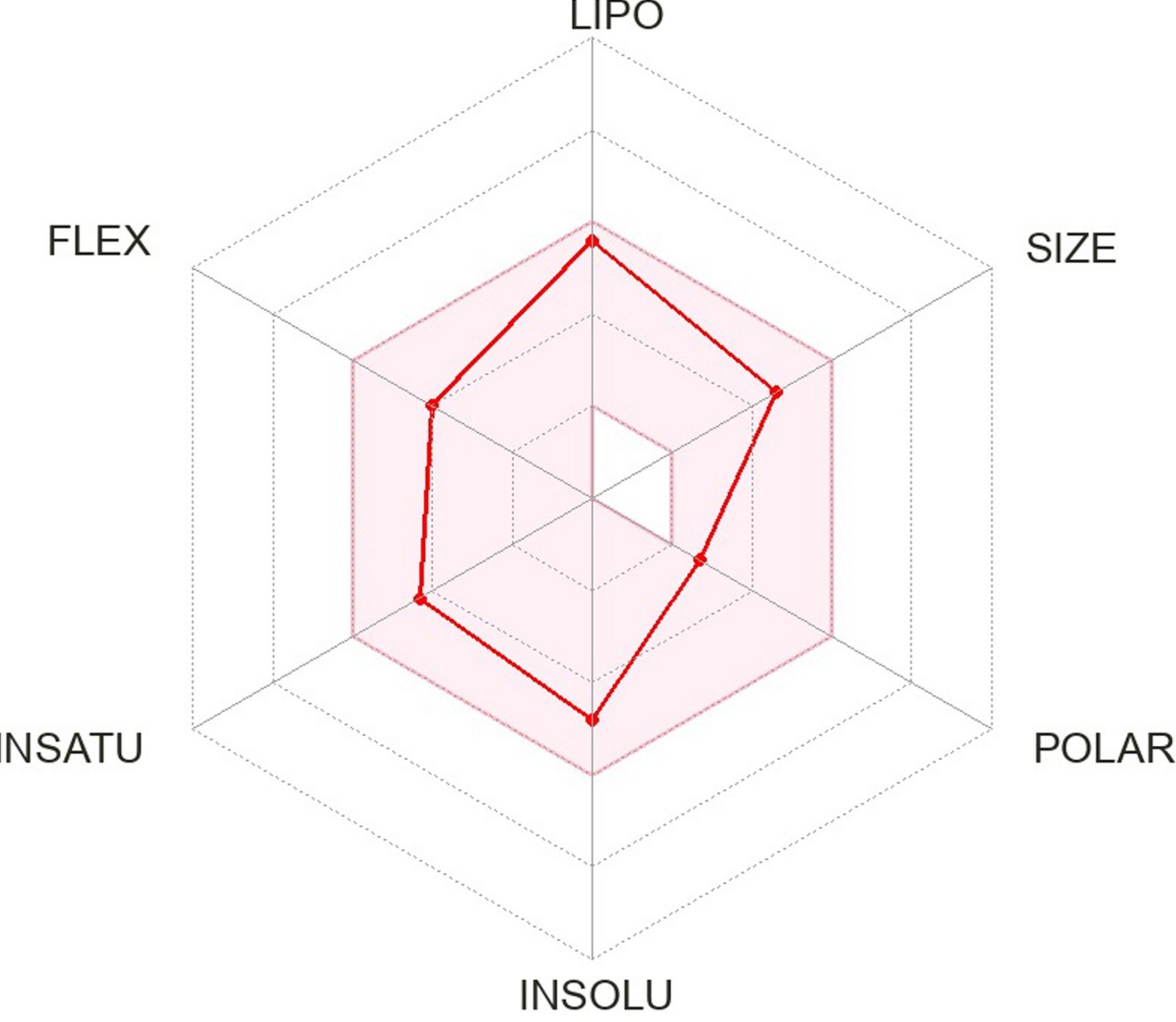

**Fig 7. Radar plots of six SSRI drugs - (1) 4184 (2) 2771 (3) 4205 (4) 5533 (5) 4543 (6) 2160 (7)3152.**

**Table 2. Binding affinity of the 24 docked complex.**

| Drug Name | CID ID | Binding Affinity |
|---|---|---|
| Mianserin | 4184 | -9 |
| Citalopram | 2771 | -8.7 |
| Mirtazapine | 4205 | -8.4 |
| Trazodone | 5533 | -8.3 |
| Nortriptyline | 4543 | -8.2 |
| Amitriptyline | 2160 | -8.2 |
| Sertraline | 68617 | -8.1 |
| Paroxetine | 43815 | -8 |
| Trimipramine | 5584 | -7.9 |
| Dosulepin | 5284550 | -7.8 |
| Reboxetine | 127151 | -7.7 |
| Imipramine | 3696 | -7.7 |
| Fluoxetine | 3386 | -7.3 |
| Escitalopram | 146570 | -7.3 |
| Duloxetine | 60835 | -7.3 |
| Clomipramine | 2801 | -7.3 |
| Agomelatine | 82148 | -7.3 |
| Isocarboxazid | 3759 | -6.8 |
| Venlafaxine | 5656 | -6.7 |
| Tranylcypromine | 5530 | -6.7 |
| Bupropion | 444 | -6.5 |
| Fluvoxamine | 5324346 | -6.3 |
| Moclobemide | 4235 | -5.5 |
| Phenelzine | 3675 | -5.3 |

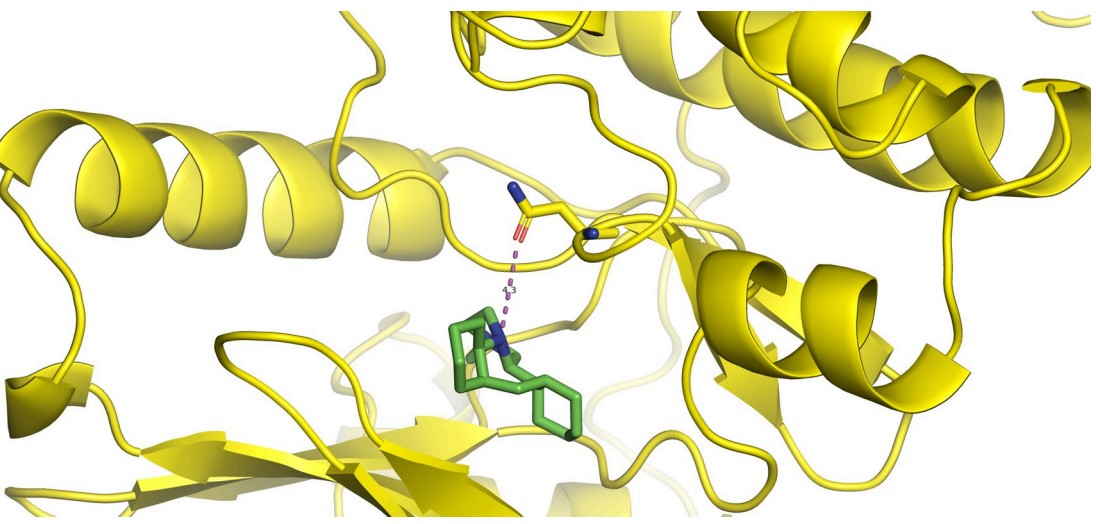

**Fig 8. Docked images of ligand-protein complex.** Ligand targeting the protein MARK4 as Fig (8) 4184 (9) 2771 (10) 4205 (11) 5533 (12) 4543 (13) 2160 (14) 3152.

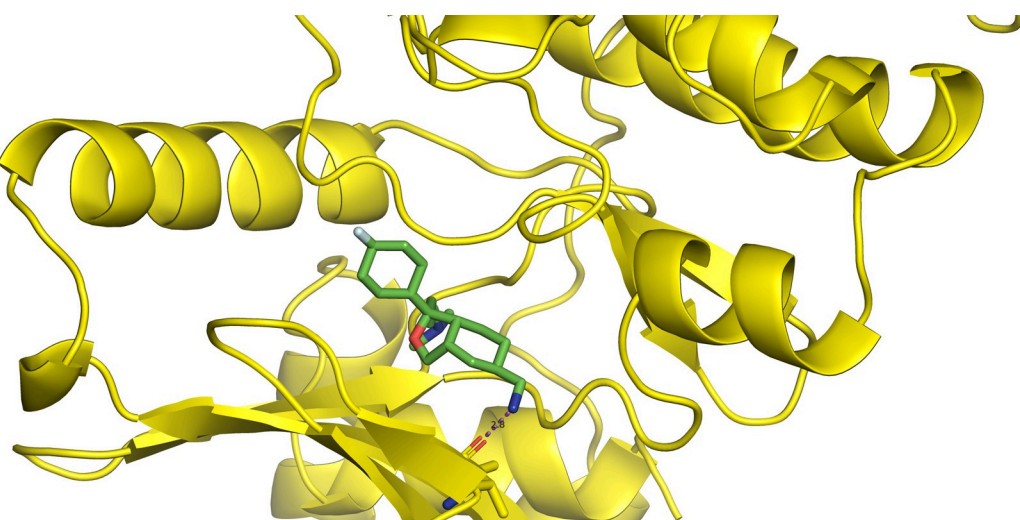

**Fig 9. Docked images of ligand-protein complex.** Ligand targeting the protein MARK4 as Fig (8) 4184 (9) 2771 (10) 4205 (11) 5533 (12) 4543 (13) 2160 (14) 3152.

Å and 3.9 Å, respectively. 5SE1-4184 formed a bond length with ASN183, which was 4.3 Å. 5SE1-4205 formed an h-bond interaction with GLU182 ASN183 with bond lengths of 2.8 Å and 3.3 Å, respectively. 5SE1-5533 formed an h-bond interaction with ASP196 with a bond length of 2.7 Å. The compound 4553 showed no h-bond interaction with the active binding site of the protein. Besides this h-bond interaction, all the compounds show salt bridges, as shown in the ligand interaction figures. Moreover, the compound 5533 additionally showed halogen interaction. These six drugs are considered further for molecular dynamics simulation, and we considered donepezil as a control, as it is an established drug for AD. The conformation of docked Sertraline, Fluoxetine, Escitalopram, Fluvoxamine, Paroxetine and Citalopram with the MARK4 is shown in the Figs 8–14. The suitable docked pose was selected by mimicking the crystal structure of the MARK4 complex with pyrazolopyrimidine inhibitor

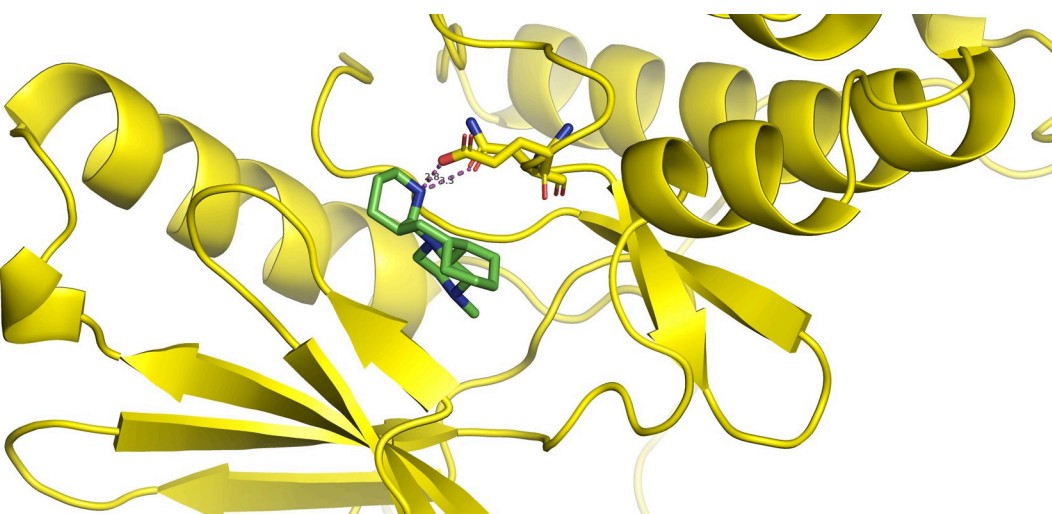

**Fig 10. Docked images of ligand-protein complex.** Ligand targeting the protein MARK4 as Fig (8) 4184 (9) 2771 (10) 4205 (11) 5533 (12) 4543 (13) 2160 (14) 3152.

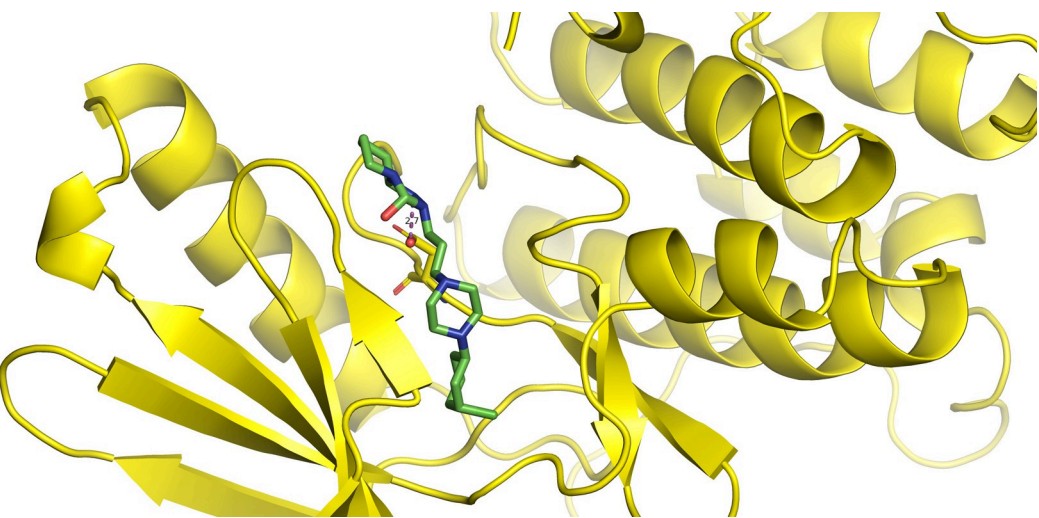

**Fig 11. Docked images of ligand-protein complex.** Ligand targeting the protein MARK4 as Fig (8) 4184 (9) 2771 (10) 4205 (11) 5533 (12) 4543 (13) 2160 (14) 3152.

(PDB: 5ES1). The docked complex where Sertraline, Fluoxetine, Escitalopram, Fluvoxamine, Paroxetine and Citalopram was present at the same position occupied by pyrazolopyrimidine inhibitor in the crystal structure, was selected for further analysis. Binding free energy of Sertraline, Fluoxetine, Escitalopram, Fluvoxamine, Paroxetine and Citalopram to the MARK4 was found to be - 9 kcal/mol,-8.7kcal/mol, -8.4kcal/mol, -8.3kcal/mol, -8.2 kcal/mol and -8.2 kcal/mol respectively. All six drugs form several close interactions to active site residues of MARK4 such as Ile62, Lys85, Val116, Met132, Tyr134, Ala135, Leu185, Ala195 and Asp196, and forming one hydrogen bonds with Lys85, and several non-covalent interactions such as Alkyl, Pi-Alkyl and Van der Waals interactions offered by the protein MARK4. It is found that all six drugs are placed at the same position where co-crystal ligand pyrazolopyrimidine inhibitor is placed and it is interacting with the same active site residues, to which co-crystal ligand

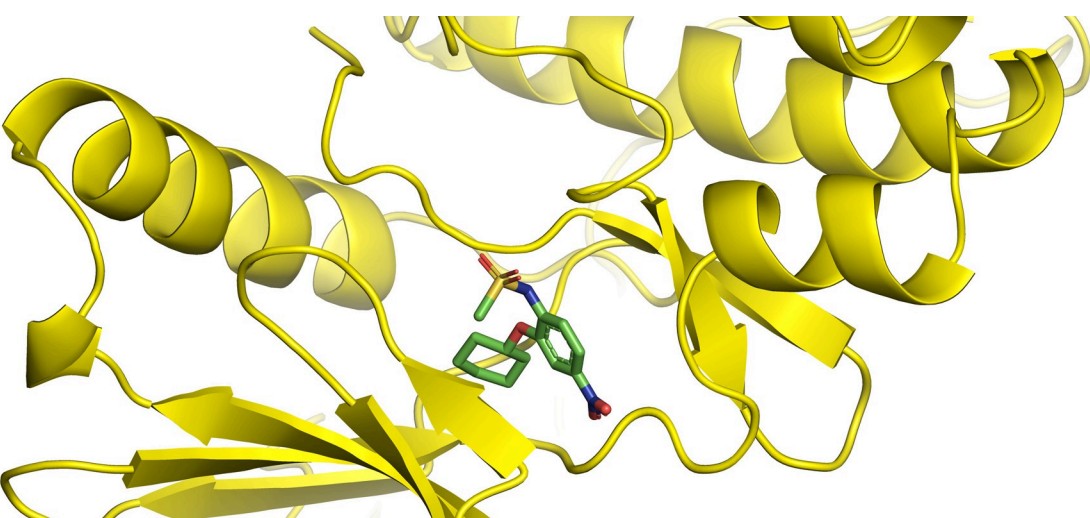

**Fig 12. Docked images of ligand-protein complex.** Ligand targeting the protein MARK4 as Fig (8) 4184 (9) 2771 (10) 4205 (11) 5533 (12) 4543 (13) 2160 (14) 3152.

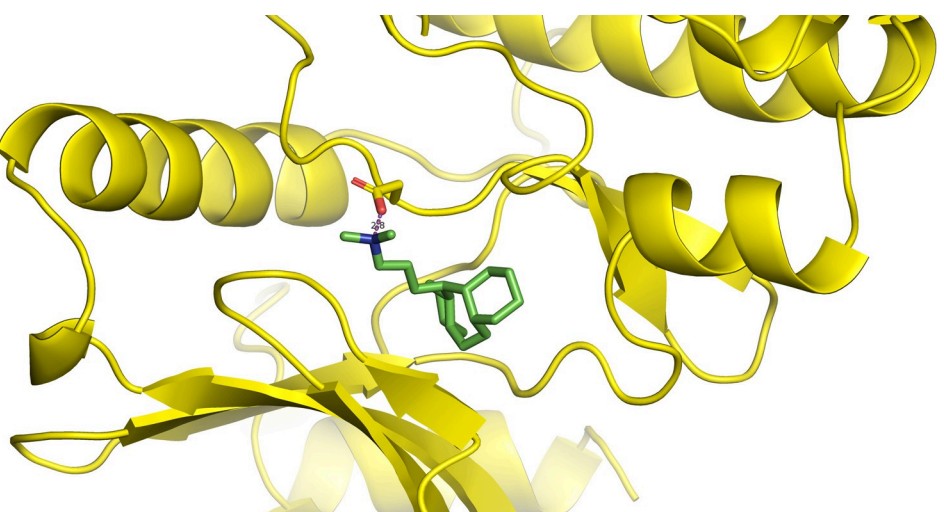

**Fig 13. Docked images of ligand-protein complex.** Ligand targeting the protein MARK4 as Fig (8) 4184 (9) 2771 (10) 4205 (11) 5533 (12) 4543 (13) 2160 (14) 3152.

pyrazolopyrimidine inhibitor is interacting with. The analysis of docked conformations clearly indicates that all six drugs bind deeper into the cavity, and perhaps decrease the accessibility of MARK4 which may be responsible modulation of its biological functions.

## 3. Molecular dynamics of 24 SSRI drugs

Based on the top six binding free energy of Sertraline, Fluoxetine, Escitalopram, Fluvoxamine, Paroxetine and Citalopram to the MARK4 which was found to be -9 kcal/mol,-8.7kcal/mol, -8.4kcal/mol, -8.3kcal/mol, -8.2 kcal/mol and -8.2 kcal/mol respectively from 24 drugs, we proceed towards further analysis which include molecular dynamics simulations at 100ns followed by MMGBSA. A molecular dynamics simulation was performed to assess the dynamic behaviour of the protein. The simulation system's stability was assessed by examining the Root

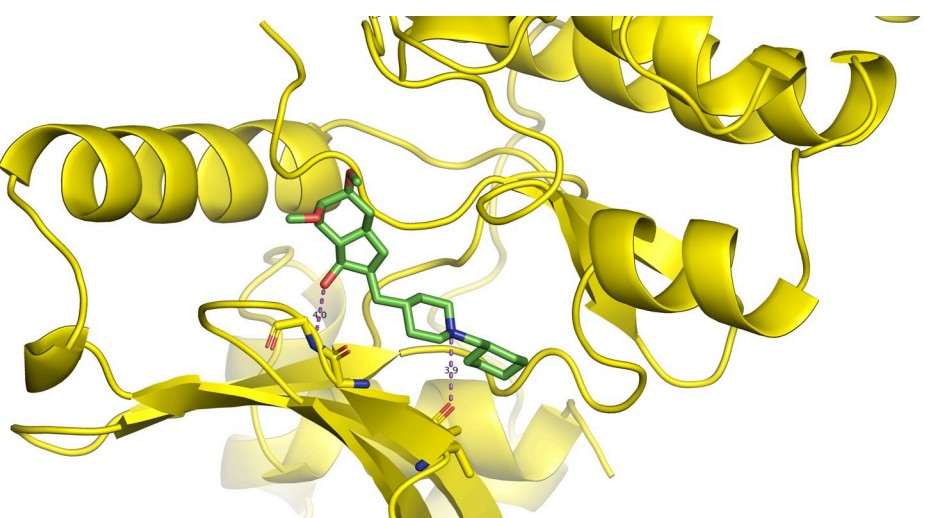

**Fig 14. Docked images of ligand-protein complex.** Ligand targeting the protein MARK4 as Fig (8) 4184 (9) 2771 (10) 4205 (11) 5533 (12) 4543 (13) 2160 (14) 3152.

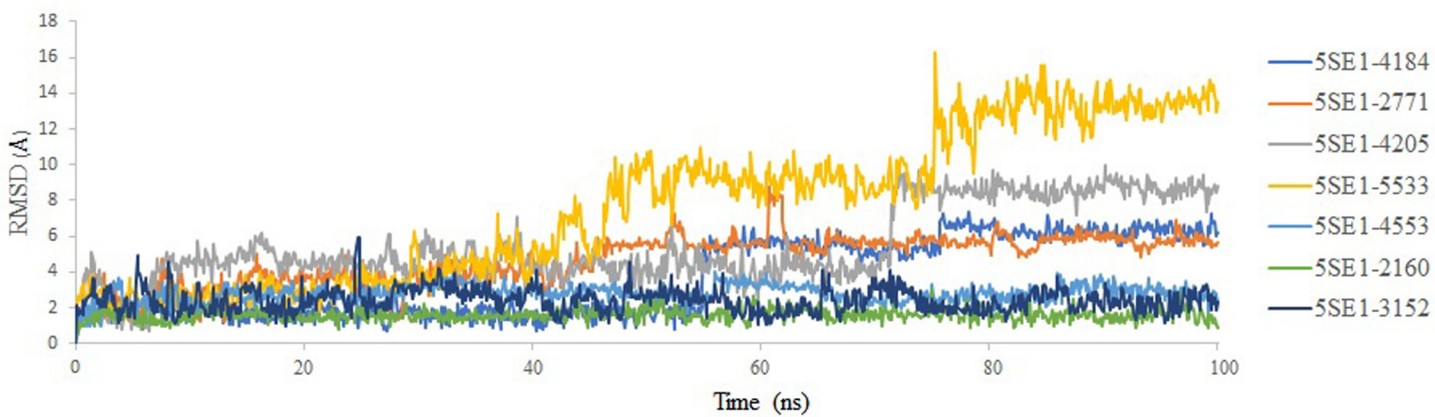

**Fig 15.** RMSD analysis of MD simulation trajectory of 5es1 in complex with (A) 4184, (B) 2771, (C) 4205, (D) 5533, (E) 4543 (F) 2160 (G) 3152.

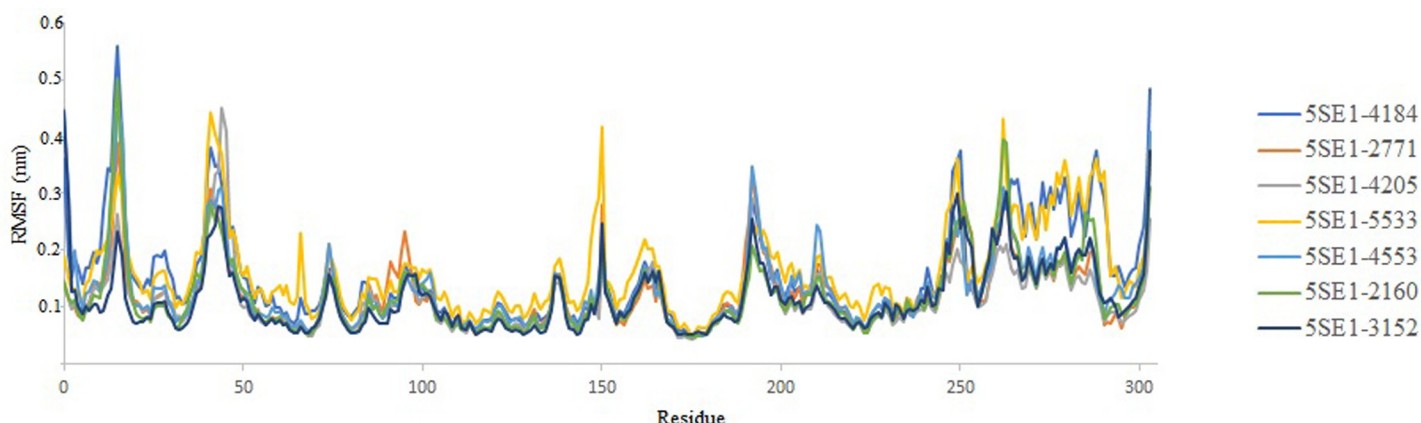

**Fig 16.** RMSF of protein (5es1) in complex with (A) 4184, (B) 2771, (C) 4205, (D) 5533, (E) 4543 (F) 2160 (G) 3152.

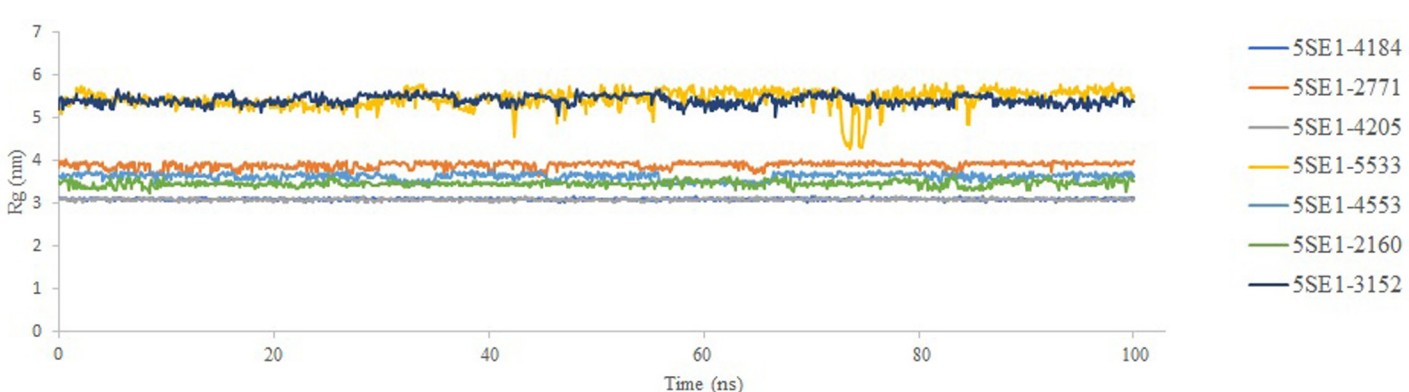

**Fig 17.** Radius of gyration of 5SE1 in complex with (A) 4184, (B) 2771, (C) 4205, (D) 5533, (E) **4553**, (F) 2160, (G) 3152.

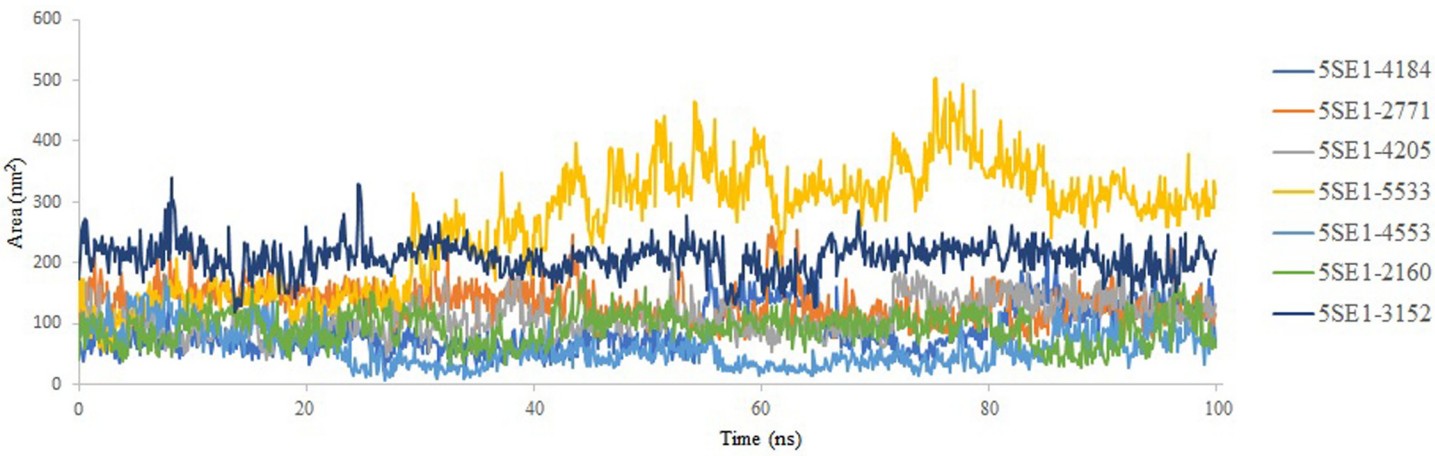

**Fig 18.** SASA of 5es1 in complex with (A) 4184, (B) 2771, (C) 4205, (D) 5533, (E) 4543 (F) 2160 (G) 3152.

Mean Square Deviations (RMSD) values throughout the trajectory. The RMSD number calculates the difference between the two structures and shows how stable the simulated system is. A lower RMSD number implies a more minor difference between the two systems. Additionally, a significant factor for examining the dynamics of the protein-ligand complexes is the stability of the simulated system, which is indicated by a less prominent variation in the RMSD value. The kinetics and the binding interactions of protein-ligand complexes can be better understood using RMSD plots, and it also sheds light on the stability and viability of these complexes as prospective therapeutic targets. Root mean square deviation RMSD plot of protein concerning ligands for 100ns MD simulation, showing most of the complexes are stable throughout the simulation trajectory, but compound 5533 showed less stability near 77–78 ns time frame; similarly, compound 4205 showed the same. The compound 4184 showed less stability at 55-56ns. The compounds 2771 and 4205, along with 5SE1, showed very stable at the

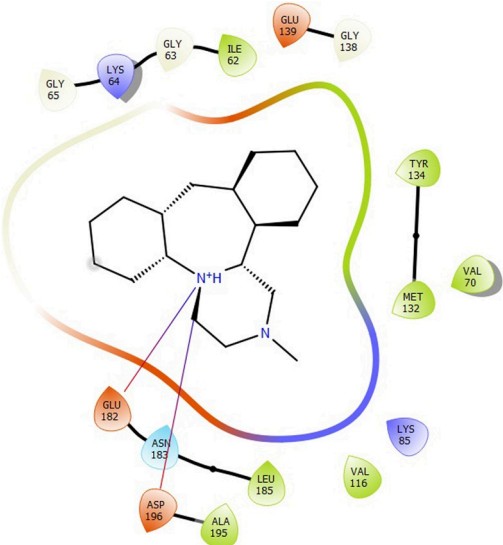

**Fig 19. Protein-ligand contact summary of the target protein MARK4 with the ligands (19) 4184, (20) 2771, (21) 4205, (22) 5533, (23) 4543 (24) 2160 (25) 3152.**

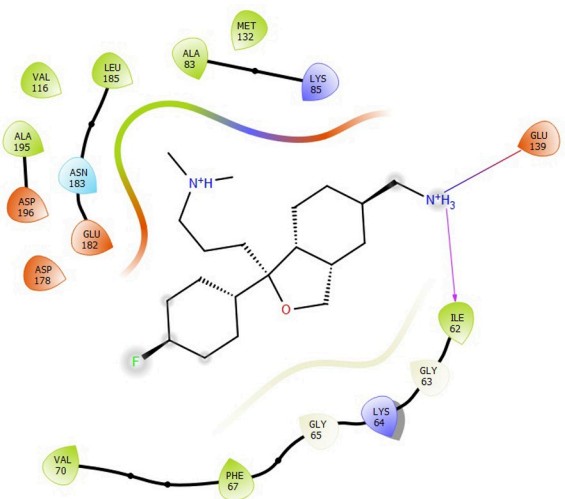

**Fig 20. Protein-ligand contact summary of the target protein MARK4 with the ligands (19) 4184, (20) 2771, (21) 4205, (22) 5533, (23) 4543 (24) 2160 (25) 3152.**

end of the simulation as in the controlled 3152 compounds. The controlled compound 3152 showed a slight instability at 25–40 ns, but after 65 ns, it regained stability and maintained it till 100 ns. The compound 4205 shows a jump in stability at 75ns, and after that, it maintains that stability till 100ns. The RMSD plot for the rest of the protein-ligand complex showed stability, but the compound 5533 is very unstable throughout the trajectory and attained an RMSD value of 14–16 Å. The compound 4184-5SE1 showed stability throughout the trajectory with RMSD value ranging between 1–2 Å. All the compounds showed stability till 45 ns properly. After that, they attained conformational changes, which formed a stronger interaction between the protein and ligand and RMSD plot is shown as Fig 15.

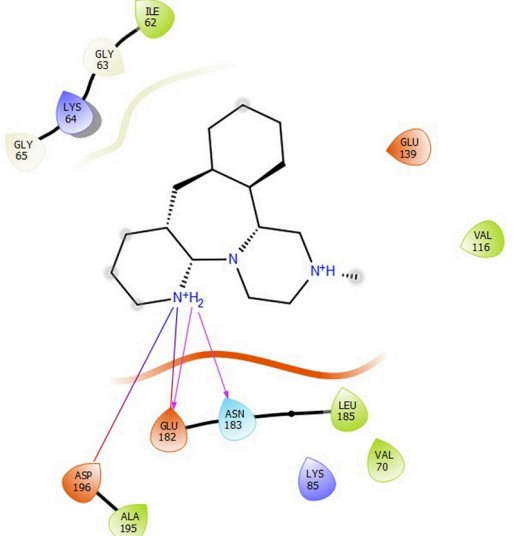

**Fig 21. Protein-ligand contact summary of the target protein MARK4 with the ligands (19) 4184, (20) 2771, (21) 4205, (22) 5533, (23) 4543 (24) 2160 (25) 3152.**

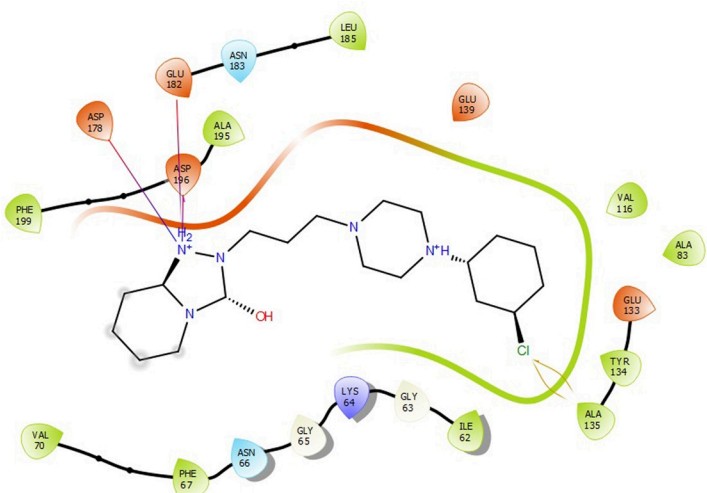

**Fig 22. Protein-ligand contact summary of the target protein MARK4 with the ligands (19) 4184, (20) 2771, (21) 4205, (22) 5533, (23) 4543 (24) 2160 (25) 3152.**

The root-mean-square-fluctuation (RMSF) is used to describe flexibility differences among residues. The backbone RMSF of each residue of the three systems, was calculated to analyse the backbone structure's flexibility. The vital amino acids between 10–25 show higher flexibility with an RMSF value of 0.5–0.6 nm. The average rmsf value is 0.35nm throughout the trajectory. The compound 2771 4205 showed the most negligible fluctuations throughout the trajectory compared to the controlled compound 3152. Amino acid residues of 5SE1: THR61, ILE62, Gly63, SER96, LEU97, GLN98, LYS99, LEU100, and PHE101 showed more flexibility in these two compounds 2160 and 4553 and their rmsf value ranging between 0.15–0.52 nm. This indicated that the conformational changes of the systems at the end of simulations were relatively small. The RMSF is shown in Fig 16.

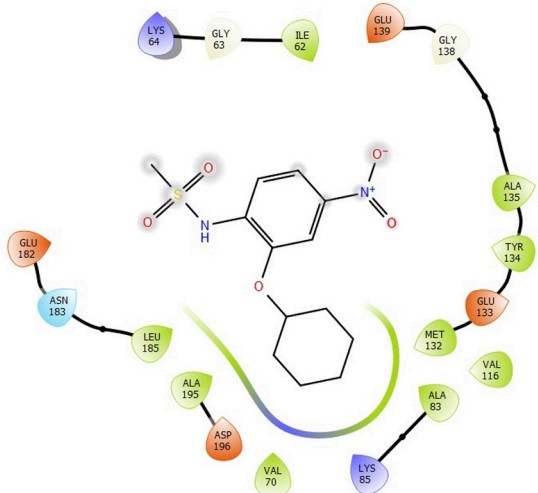

**Fig 23. Protein-ligand contact summary of the target protein MARK4 with the ligands (19) 4184, (20) 2771, (21) 4205, (22) 5533, (23) 4543 (24) 2160 (25) 3152.**

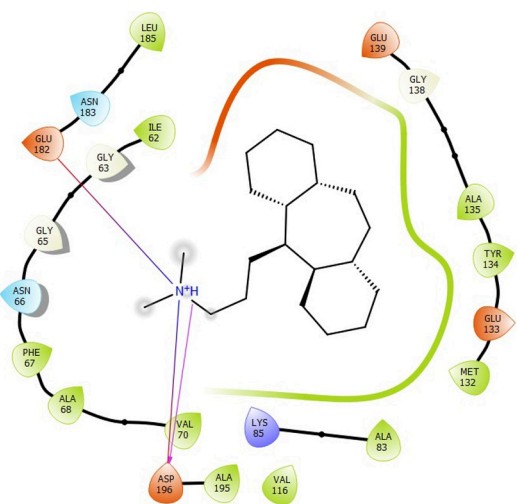

**Fig 24. Protein-ligand contact summary of the target protein MARK4 with the ligands (19) 4184, (20) 2771, (21) 4205, (22) 5533, (23) 4543 (24) 2160 (25) 3152.**

A protein's volume and shape are described by the radius of gyration (Rg), which offers information on the compactness and stability of the protein's structure. A higher Rg value indicates an expanded system, which suggests a less dense and compact protein structure. A stable folded protein, on the other hand, usually shows a constant Rg value. Changes in the Rg value can signal structural changes in the protein, such as unfolding or denaturation, which can happen under various circumstances or due to various external factors. After analysing the plot, it was observed that the compactness of the compound 2771, 4205 is much less compared to the controlled 3152 with Rg value ranges between 3.2–3.75nm. This indicates that the compound 2771 and 4205 are better compact than 3152. The rest of the compounds showed

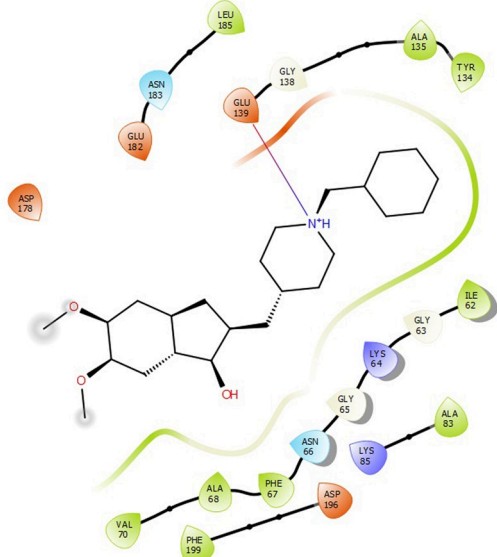

**Fig 25. Protein-ligand contact summary of the target protein MARK4 with the ligands (19) 4184, (20) 2771, (21) 4205, (22) 5533, (23) 4543 (24) 2160 (25) 3152.**

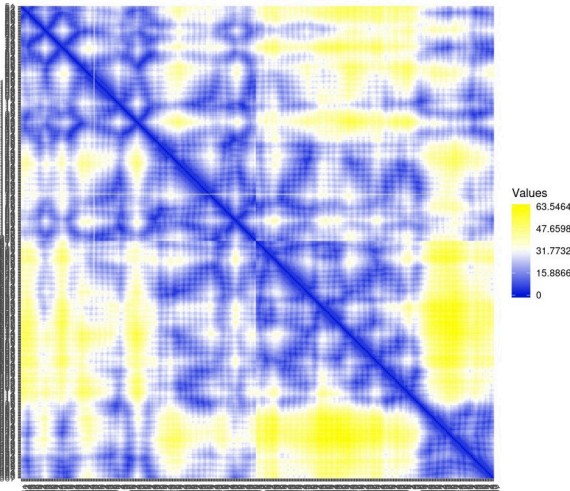

**Fig 26. Residue contact map of 5es1 in complex with (26) 4184, (27) 2771, (28) 4205, (29) 5533, (30) 4543 (31) 2160 (32) 3152.**

compactness values ranging between 3-4nm, whereas the compound 5533 showed very similar compactness values to 3152 and all the results are shown in Fig 17.

The solvent-accessible surface area (SASA) plot analysis of the molecular dynamic simulation was used to gauge the protein's environmental exposure over time. The plot revealed information about the dynamics and structural changes of the protein during the simulation and is shown in Fig 18. It was feasible to pinpoint areas of the protein with varying solvent exposure by keeping track of changes in SASA levels. This could point to protein-ligand interactions, structural rearrangements, or solvent-shielding effects. The SASA plot analysis helped to understand the behaviour of the protein better and offered vital data for researching its relationships and function in a dynamic context. In this study, after analysing solvent-accessible

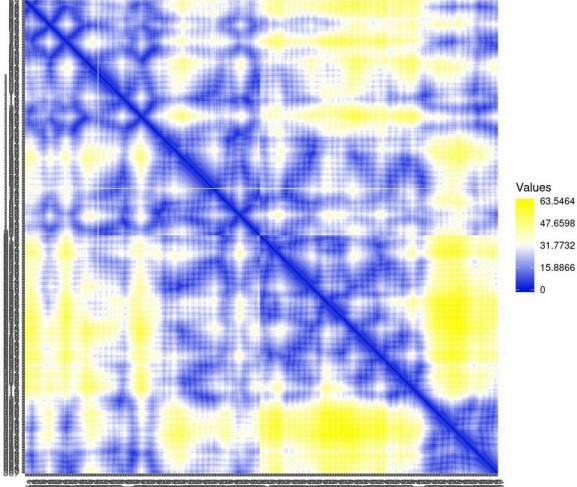

**Fig 27. Residue contact map of 5es1 in complex with (26) 4184, (27) 2771, (28) 4205, (29) 5533, (30) 4543 (31) 2160 (32) 3152.**

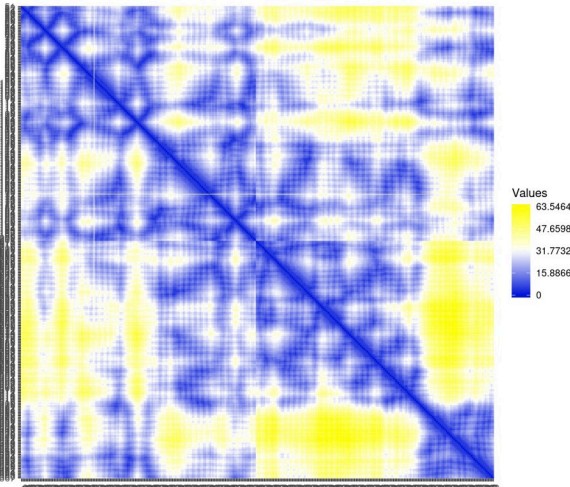

**Fig 28. Residue contact map of 5es1 in complex with (26) 4184, (27) 2771, (28) 4205, (29) 5533, (30) 4543 (31) 2160 (32) 3152.**

surface area for all the proteins and the compounds throughout the simulation, it was observed that all the systems have SASA values ranging between 53.96 nm2 and 302.11 nm2. Compounds 2771 and 4205 showed excellent solvent accessibility with 54–150 nm2 values. Overall, it can be concluded from this, which is comparatively reasonable compared to the controlled 3152 proteins, and the compound has excellent accessibility to the solvent environment.

The hydrogen bond interaction between the protein and ligand changes throughout the trajectory at 0ns, there are approx. 1–2 h-bond, and the interaction increases near 17–18 ns due to some conformational changes of the compounds along with the conformational changes of the binding site residues. After 45ns, the interaction starts falling at 100ns; it maintained 2–3 interactions. The hydrogen bond interaction is shown in S1 Fig. The ligand interaction with the protein's binding site is shown in Figs 19–25. In all these images, it is observed that most of

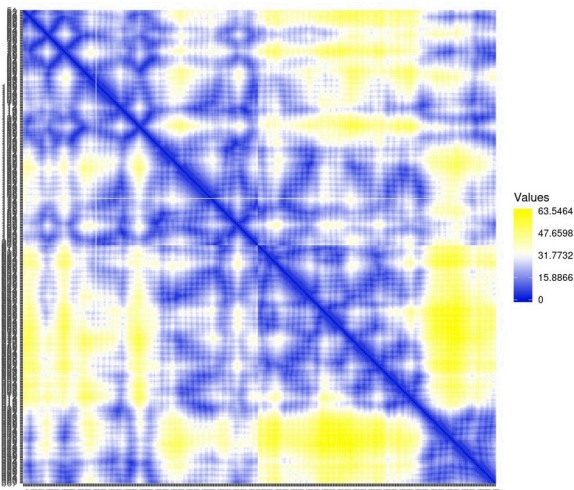

**Fig 29. Residue contact map of 5es1 in complex with (26) 4184, (27) 2771, (28) 4205, (29) 5533, (30) 4543 (31) 2160 (32) 3152.**

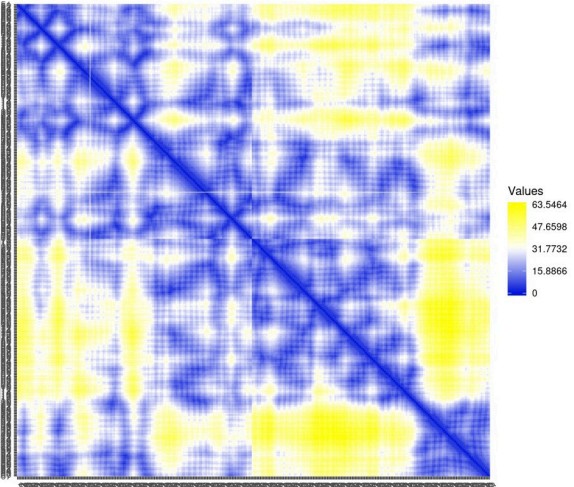

**Fig 30. Residue contact map of 5es1 in complex with (26) 4184, (27) 2771, (28) 4205, (29) 5533, (30) 4543 (31) 2160 (32) 3152.**

the compounds form h-bonds with the protein binding site and also show salt bridges. The compound 5533 also showed halogen bond interaction with the Ala135 residue of the protein's binding site.

Residue contact map and the mean smallest distance between C-alpha atoms of each amino acid residue 5SE1 corresponding to docked compounds. The residue contact map is analysed to calculate the smallest distance between c-alpha atoms that unify protein-ligand docked complexes that influence secondary structure elements to know their allosteric effects on the protein. The protein-ligand complex 5SE1-2771 5SE1-4205 showed atomic distance as shown in the above plots. From the above comparative analysis, it is evident that the interacting amino acids of the complexes were near form strong binding and stable conformations of the ligands towards the active pocket of 5SE1 and it is shown in Figs 26–32.

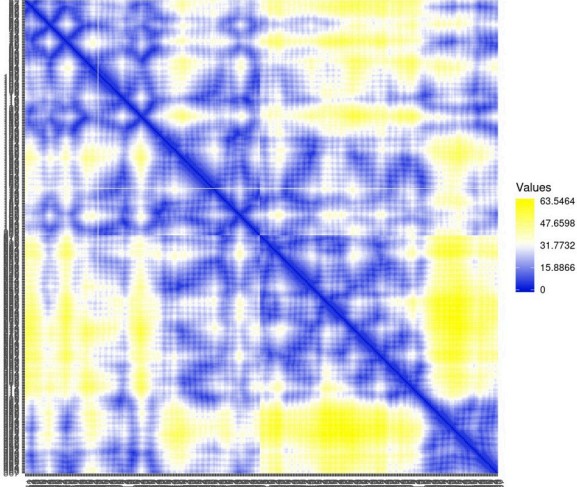

**Fig 31. Residue contact map of 5es1 in complex with (26) 4184, (27) 2771, (28) 4205, (29) 5533, (30) 4543 (31) 2160 (32) 3152.**

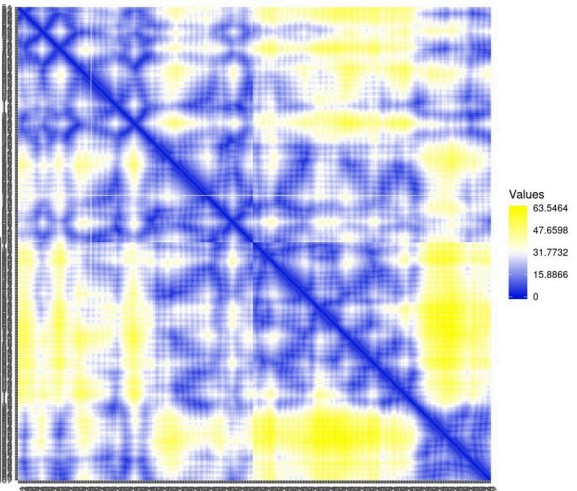

**Fig 32. Residue contact map of 5es1 in complex with (26) 4184, (27) 2771, (28) 4205, (29) 5533, (30) 4543 (31) 2160 (32) 3152.**

The Free Energy Landscape (FEL) displayed an intricate energy landscape with several basins and transition states, pointing to a system with a rich conformational diversity. Numerous shallow wells for the compounds with protein signify metastable states, whereas substantial barriers signify transitions that require much energy. Notably, the conformation with the lowest energy is the most stable. According to this FEL study, the system investigates several structural configurations linked to a different energy level. Understanding the system's behaviour, including possible binding paths, conformational changes, and the thermodynamics driving molecular dynamics, is essential for comprehending biological processes and logical drug design. These insights into the energy landscape give vital information about the system's behaviour. After analysing the FEL plot, it was observed that the FEL achieved global minima of C-alpha backbone atoms of proteins concerning RMSD and rGyr. The FEL forecast for

Free Energy Landscape

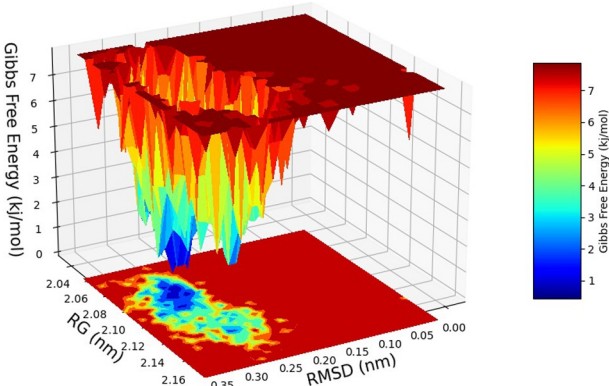

**Fig 33. Free energy landscape of 5es1 in complex with (33) 4184, (34) 2771, (35) 4205, (36) 5533, (37) 4543 (38) 2160 (39) 3152.**

Free Energy Landscape

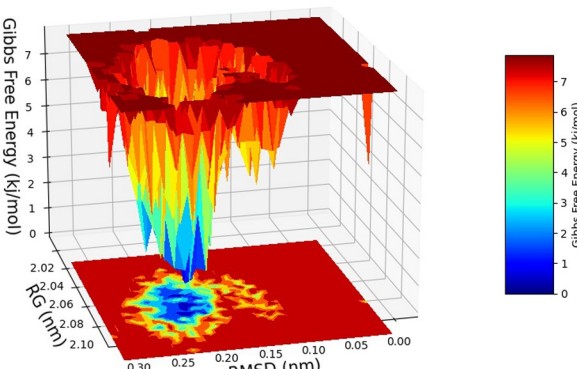

**Fig 34. Free energy landscape of 5es1 in complex with (33) 4184, (34) 2771, (35) 4205, (36) 5533, (37) 4543 (38) 2160 (39) 3152.**

deterministic behaviour of 5SE1 to lowest energy state owing to its high stability and best conformation at 2771 and 4205 bound state compared to the controlled 3152 bound state. The result of FEL is shown in Figs 33–39.

Principal Component investigation (PCA) was used to examine the MD simulations of the 5SE1 protein and certain chemicals' trajectories. The main aim was to understand atoms' complex, erratic motions in amino acid residues. This procedure revealed instances when the protein structure suffered deformation due to greater flexibility, shedding light on the variety in trajectories. This was accomplished by tracking the motion of internal coordinates in a three-dimensional space, recording data throughout 100 nanoseconds, and presenting the data in a scatter pattern. Later, the idea of orthogonal sets or Eigenvectors was used to understand the controlled motion of each trajectory. In particular, the PCA analysis revealed an intriguing

Free Energy Landscape

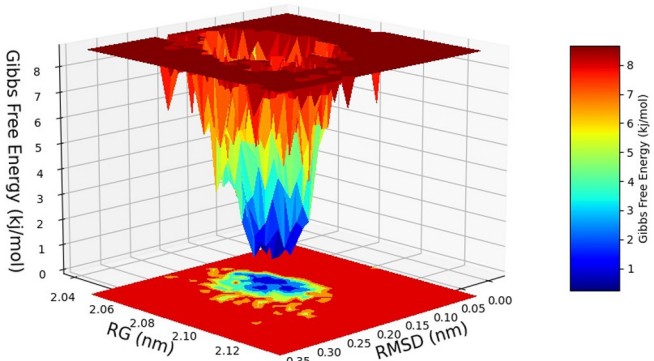

**Fig 35. Free energy landscape of 5es1 in complex with (33) 4184, (34) 2771, (35) 4205, (36) 5533, (37) 4543 (38) 2160 (39) 3152.**

Free Energy Landscape

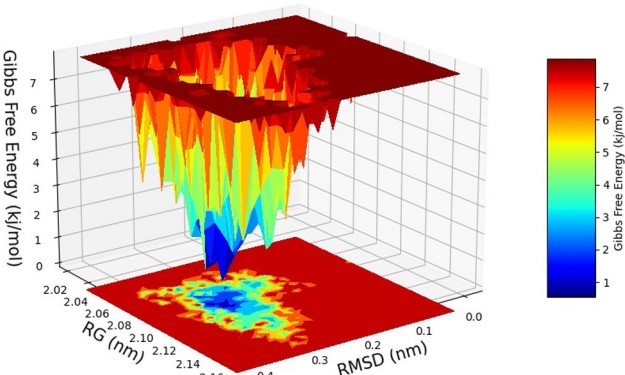

**Fig 36. Free energy landscape of 5es1 in complex with (33) 4184, (34) 2771, (35) 4205, (36) 5533, (37) 4543 (38) 2160 (39) 3152.**

finding: when one advances into higher modes, especially approaching PC4. The result of PCA is shown in Figs 40–46.

After analysing all the molecular docking results using PyMOL, MD Simulation trajectory analysis, it can be concluded that the citalopram and mirtazapine showed the best results in all aspects as we compared them with control 3152. It was observed that during simulation for 100ns, binding of these compounds to the active site of protein caused some conformational changes of the residues of the protein, due to which, from a reported time frame, it obtained stability till 100ns and binds appropriately as compared to earlier.

Free Energy Landscape

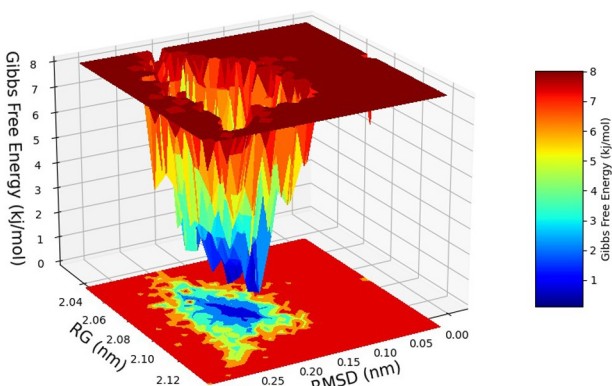

**Fig 37. Free energy landscape of 5es1 in complex with (33) 4184, (34) 2771, (35) 4205, (36) 5533, (37) 4543 (38) 2160 (39) 3152.**

Free Energy Landscape

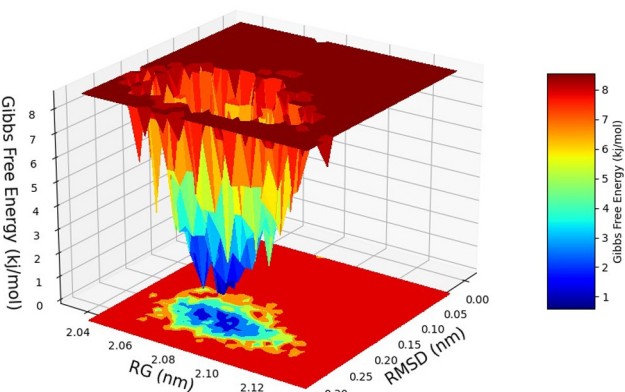

**Fig 38. Free energy landscape of 5es1 in complex with (33) 4184, (34) 2771, (35) 4205, (36) 5533, (37) 4543 (38) 2160 (39) 3152.**

## 4. Molecular Mechanics Generalised Born Surface Area (MMGBSA) calculations

Further protein-ligand binding affinity was validated by calculating the complexes' MMGBSA binding free energies.MM/GBSA is emerging as a valuable and practical approach to predicting the binding energy. The Prime MM/GBSA module of the Schrodinger Suite was used to calculate the binding free energy of receptor-ligand complexes. After analysing the MMGBSA plot, it was observed that for most of the protein-ligand complexes, the values ranged between -40-50 region as shown in orange colour in the plot. Based on the MMGBSA result we can observe that the binding value for both 2771 and 4205 is between -30–40 which is very similar to the 5SE1-3152 controlled compound whose MMGBSA binding value also ranged between

Free Energy Landscape

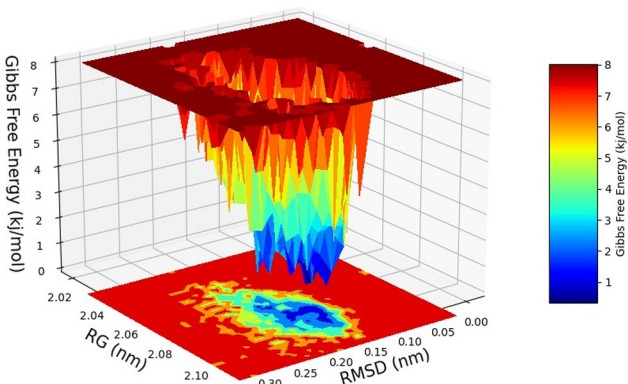

**Fig 39. Free energy landscape of 5es1 in complex with (33) 4184, (34) 2771, (35) 4205, (36) 5533, (37) 4543 (38) 2160 (39) 3152.**

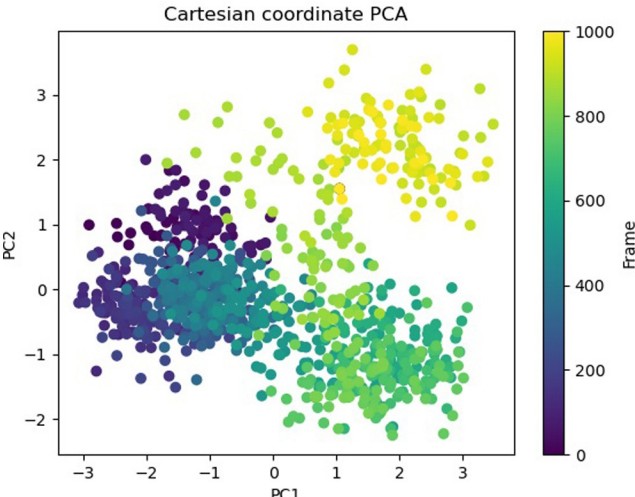

**Fig 40. PCA of 5es1 in complex with (40) 4184, (41) 2771, (42) 4205, (43) 5533, (44) 4543 (45) 2160 (46) 3152.**

-30-40, respectively. The binding value for 4184, 5533, 4543 and 2160 ranged between -40-50 which is not close to the control drug (donepezil). The MMGBSA result is shown in Fig 47.

## Discussion

Molecular docking is a widely recognised technique in the field of structure-based drug design, enabling the examination of interactions between tiny molecules and molecular targets. By employing this methodology, a more comprehensive comprehension of the behaviour of small molecules within certain protein targets can be achieved, facilitating the ability to forecast the structure of ligand-receptor complexes. This capability holds significant importance in the realm of pharmaceutical research. The repurposing of pharmaceuticals has consistently demonstrated efficacy in the treatment of many medical issues and illnesses. SSRIs have demonstrated favourable outcomes in the treatment of AD. The present study employed molecular

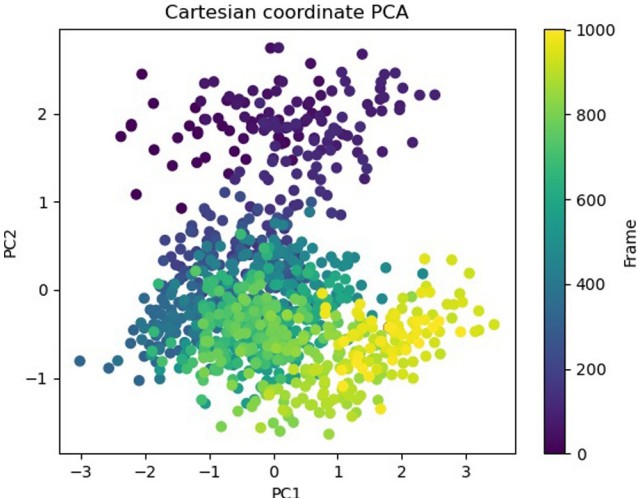

**Fig 41. PCA of 5es1 in complex with (40) 4184, (41) 2771, (42) 4205, (43) 5533, (44) 4543 (45) 2160 (46) 3152.**

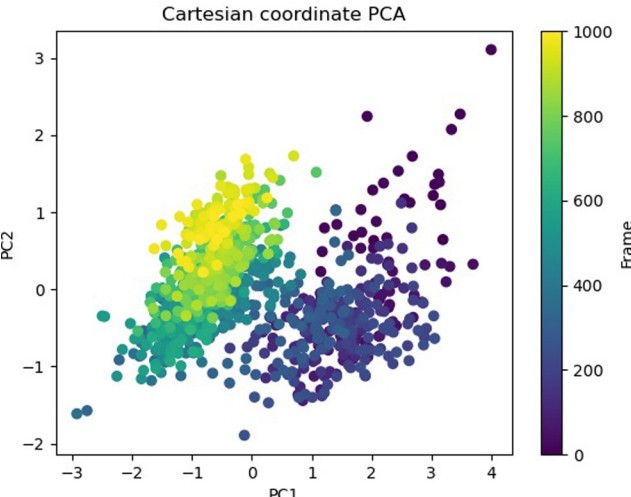

**Fig 42. PCA of 5es1 in complex with (40) 4184, (41) 2771, (42) 4205, (43) 5533, (44) 4543 (45) 2160 (46) 3152.**

docking and subsequent molecular dynamics simulations to assess the interaction between SSRI medications and MARK4 protein. The objective was to identify the most favourable SSRI drug candidate capable of selectively targeting the overexpressed MARK4 protein in AD. The MMGBSA method was employed to calculate the binding energy of the top six medications, taking into account their binding affinity. This approach was utilised to verify the docking results. The results of the analysis indicated that these pharmaceuticals consistently and reliably interacted with the active site residues of MARK4 over the whole simulation trajectory. Mirtazapine, a pharmaceutical compound that specifically targets the MARK4 enzyme, exhibits potential for further exploration in various research settings, including *in-vitro*, *in-vivo*, and clinical investigations pertaining to AD. Previous studies conducted by Shamsi et al. in 2020 have demonstrated the binding affinity of donepezil, rivastigmine tartrate, and other cholinergic inhibitors to the active site of the MARK4 enzyme [27]. According to a recent study conducted by Waseem et al. in 2021, it has been observed that the protein irisin, which plays a

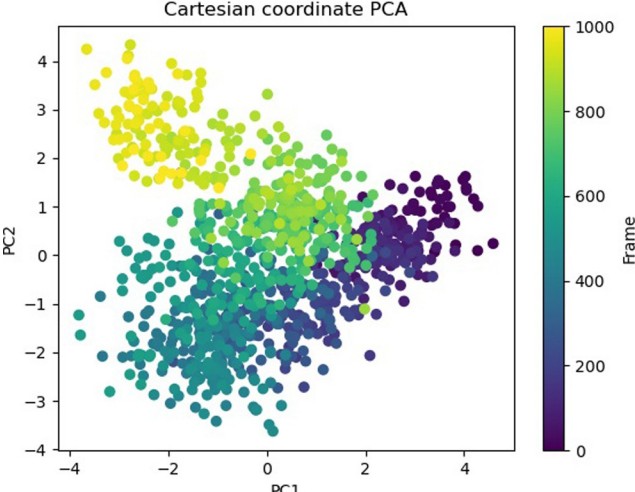

**Fig 43. PCA of 5es1 in complex with (40) 4184, (41) 2771, (42) 4205, (43) 5533, (44) 4543 (45) 2160 (46) 3152.**

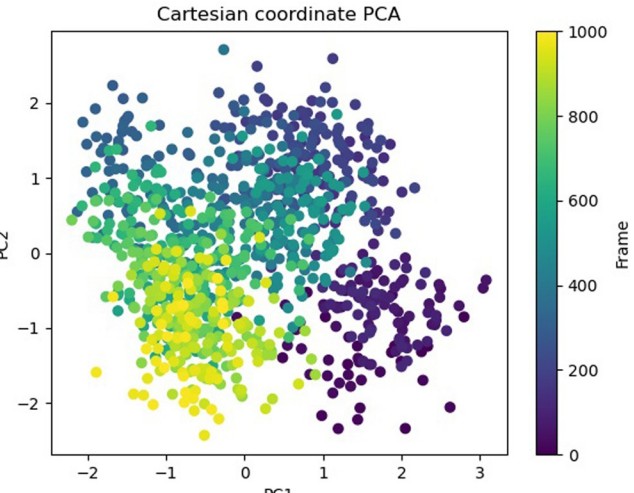

**Fig 44. PCA of 5es1 in complex with (40) 4184, (41) 2771, (42) 4205, (43) 5533, (44) 4543 (45) 2160 (46) 3152.**

critical role in the decline of memory associated with AD, has a binding affinity towards MARK4 and contributes to the preservation of its stability. The user has provided a numerical reference [28]. According to recent investigations [29], the molecules ganoderic acid A and ganoderenic acid B, produced from Ganoderma lucidium, have shown significant potential in their interaction with MARK4. According to the latest investigation, galantamine has been identified as a potentially effective inhibitor for MARK 4 [30]. Based on our investigation, bolstered by previous research, it has been determined that mirtazapine, which exhibits affinity for MARK4, is regarded as the most auspicious SSRI medication among the 24 SSRI medications under consideration. Prior to conducting clinical trials, it is imperative to validate our findings through *in vitro* and *in vivo* research.

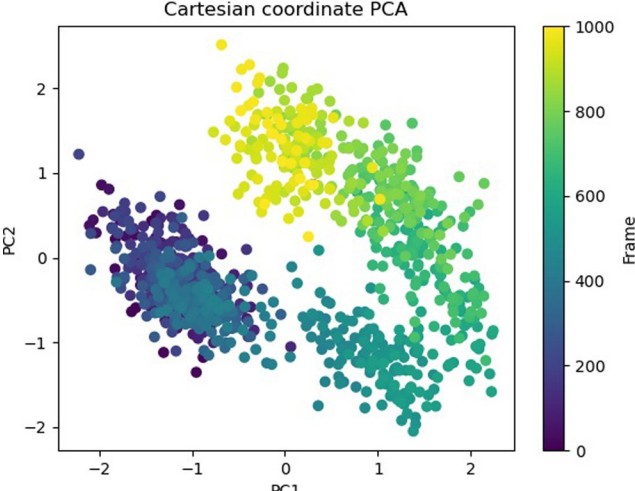

**Fig 45. PCA of 5es1 in complex with (40) 4184, (41) 2771, (42) 4205, (43) 5533, (44) 4543 (45) 2160 (46) 3152.**

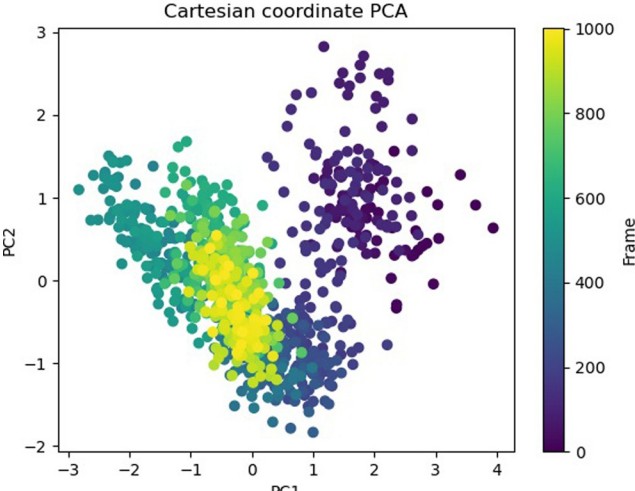

**Fig 46. PCA of 5es1 in complex with (40) 4184, (41) 2771, (42) 4205, (43) 5533, (44) 4543 (45) 2160 (46) 3152.**

## Conclusion

Currently, there are just five medications for AD that have been approved and SSRIs are used as off-label drugs in AD. In this study we presents a novel prediction that citalopram and mirtazapine are the most promising SSRIs from the list of 24 drugs which is targeting the MARK4 protein, which is related with AD. However, further validation is required through rigorous *in vitro* and *in vivo* investigations.

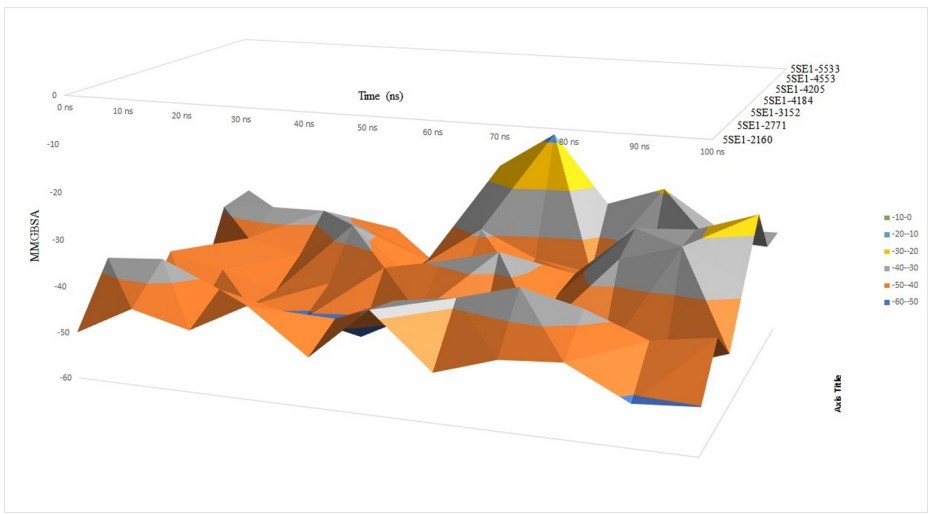

**Fig 47. MMGBSA analysis of 5es1 with the ligands (A) 4184, (B) 2771, (C) 4205, (D) 5533, (E) 4543 (F) 2160 (G) 3152.**

## Supporting information

**S1 Fig.** H-bond occupancy plot of the protein 5SE1 in complex with (A) 4184, (B) 2771, (C) 4205, (D) 5533, (E) 4543 (F) 2160 (G) 3152.
(DOCX)

**S1 Table.**
(DOCX)

**S1 Dataset. Minimal data set definition.**
(DOCX)

## Acknowledgments

I, Dr. S Rehan Ahmad, and Md. Zeyaullah would like to express our sincere gratitude to our esteemed co-authors for their invaluable scientific contributions to this project. Without their dedication, expertise, and collaborative spirit, the accomplishment of this endeavour would not have been possible. Each co-author has brought unique insights, skills, and perspectives that have enriched the research process and outcomes. Furthermore, we extend our heartfelt appreciation to the authorities of the concerned colleges and universities for their support and assistance throughout the duration of this project. This research was made possible by the collective efforts and unwavering commitment of all involved researchers, and we are deeply thankful for their contributions.

## Author Contributions

**Conceptualization:** S. Rehan Ahmad, Md. Zeyaullah.

**Data curation:** S. Rehan Ahmad, Md. Zeyaullah, Abdelrhman A. G. Altijani.

**Formal analysis:** S. Rehan Ahmad, Md. Zeyaullah, Haroon Ali.

**Funding acquisition:** Md. Zeyaullah.

**Investigation:** S. Rehan Ahmad, Md. Zeyaullah.

**Methodology:** S. Rehan Ahmad, Md. Zeyaullah.

**Project administration:** S. Rehan Ahmad, Md. Zeyaullah.

**Resources:** S. Rehan Ahmad, Md. Zeyaullah, Abdullah M. AlShahrani, Khursheed Muzammil, Faheem Ahmed.

**Software:** S. Rehan Ahmad, Md. Zeyaullah, Mohammad Suhail Khan.

**Supervision:** S. Rehan Ahmad, Md. Zeyaullah, Adam Dawria.

**Validation:** S. Rehan Ahmad, Md. Zeyaullah, Abdullah M. AlShahrani, Faheem Ahmed, Haroon Ali.

**Visualization:** S. Rehan Ahmad, Md. Zeyaullah, Mohammad Suhail Khan, Khursheed Muzammil, Ali Mohieldin.

**Writing – original draft:** S. Rehan Ahmad, Md. Zeyaullah.

**Writing – review & editing:** Abdullah M. AlShahrani, Mohammad Suhail Khan, Khursheed Muzammil, Faheem Ahmed, Adam Dawria, Ali Mohieldin, Haroon Ali, Abdelrhman A. G. Altijani.

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
