## [Decision Letter · Decision Letter 0]

19 Jan 2024

PONE-D-23-35218Exploring a Promising Anti-Depressant Drug Targeting Microtubule Affinity Receptor Kinase 4 Inhibition in Alzheimer's Disease through Molecular Docking and Dynamics SimulationPLOS ONE

Dear Dr. Ahmad,

Thank you for submitting your manuscript to PLOS ONE. After careful consideration, we feel that it has merit but does not fully meet PLOS ONE’s publication criteria as it currently stands. Therefore, we invite you to submit a revised version of the manuscript that addresses the points raised during the review process.

**The manuscript overall quality is good but still needs major changes as suggested by the reviewer. **

We look forward to receiving your revised manuscript.

Kind regards,

Sajjad Ahmad

Academic Editor

PLOS ONE

“Deanship of Scientific Research at King Khalid University, Kingdom of Saudi Arabia ,

Research Project Grant Number RGP2/355/44.”

5. Please amend either the title on the online submission form (via Edit Submission) or the title in the manuscript so that they are identical.

6. Please amend either the abstract on the online submission form (via Edit Submission) or the abstract in the manuscript so that they are identical.

7. Please upload a copy of Supplementary 1. which you refer to in your text on page 15.

Additional Editor Comments:

The reviewer raised some concerns on the methodology of the manuscript. However, i still think the paper has the potential to be published. I will give one chance to the authors to improve the manuscript in light of the reviewer comments and send the manuscript for second round of revisions after incorporating all the changes.

Reviewers' comments:

Reviewer's Responses to Questions

**Comments to the Author**

1. Is the manuscript technically sound, and do the data support the conclusions?

Reviewer #1: No

2. Has the statistical analysis been performed appropriately and rigorously? 

Reviewer #1: No

3. Have the authors made all data underlying the findings in their manuscript fully available?

Reviewer #1: No

4. Is the manuscript presented in an intelligible fashion and written in standard English?

Reviewer #1: Yes

5. Review Comments to the Author

Reviewer #1: Overall, the study presents a well-structured and comprehensive investigation into the potential of antidepressants as a treatment for Alzheimer's disease (AD). The authors have clearly identified a need for more effective AD therapies and have explored a promising therapeutic target, MARK4, which is involved in both AD and cancer progression. The use of molecular docking, molecular dynamics simulation, and MMGBSA to assess the binding affinity and potential efficacy of various antidepressants is a rigorous approach that provides valuable insights.

However, there are a few areas which need to be addressed

The selection of 24 antidepressants for this study warrants further explanation. The authors rely heavily on a 2013 study that itself drew from outdated references, raising concerns about the study's novelty and the relevance of the chosen antidepressants. The 2013 study, which served as the primary basis for selecting the antidepressants, appears to have cherry-picked compounds based on limited evidence and without considering more recent advancements in antidepressant research. Given the significant advancements in antidepressant development over the past decade, it is surprising that the authors chose to exclusively rely on compounds identified in a 2013 study.

Moreover, the authors should conduct additional literature reviews to identify newer and more relevant evidence supporting their choice of antidepressants. This could involve examining recent studies that have investigated the potential of antidepressants in treating Alzheimer's disease or exploring the mechanisms by which these compounds interact with MARK4.

The authors' reliance on ADMET analysis to narrow down the pool of potential antidepressants is also a significant concern. While ADMET studies provide valuable insights into the pharmacokinetic and toxicological properties of compounds, they should not be used solely as exclusionary criteria to filter out potential drug candidates.

The authors' reliance on molecular docking to identify promising antidepressants warrants further scrutiny. While molecular docking provides a valuable starting point for identifying potential drug interactions, it is essential to carefully validate the results to ensure their robustness and reproducibility. The authors should compare the predicted binding modes from docking with those obtained from X-ray crystallography as reported in PDB. This will help to establish the accuracy of the docking simulations and ensure that the proposed binding interactions are biologically plausible.

Justifiy in details the criteria for selecting the top six antidepressants for molecular dynamics simulation and MMGBSA.

Provide a more detailed comparison of the molecular dynamics and MMGBSA results for the six antidepressants.

6. PLOS authors have the option to publish the peer review history of their article (what does this mean?). If published, this will include your full peer review and any attached files.

Reviewer #1: **Yes: **Shafi Ullah Khan

---

## [Author Response · Author response to Decision Letter 0]

8 Feb 2024

Comments to the Author

1. Is the manuscript technically sound, and do the data support the conclusions?

Reviewer #1: No

2. Has the statistical analysis been performed appropriately and rigorously?

Reviewer #1: No

3. Have the authors made all data underlying the findings in their manuscript fully available?

Reviewer #1: No

4. Is the manuscript presented in an intelligible fashion and written in standard English?

Reviewer #1: Yes

5. Review Comments to the Author

Reviewer #1: Overall, the study presents a well-structured and comprehensive investigation into the potential of antidepressants as a treatment for Alzheimer's disease (AD). The authors have clearly identified a need for more effective AD therapies and have explored a promising therapeutic target, MARK4, which is involved in both AD and cancer progression. The use of molecular docking, molecular dynamics simulation, and MMGBSA to assess the binding affinity and potential efficacy of various antidepressants is a rigorous approach that provides valuable insights.

However, there are a few areas which need to be addressed

We would like to thanks the reviewer for spending value time and effort in reviewing the manuscript. We modified the manuscript based on the comment and we also modified conclusion to have better clarity and removed extra MMGBSA figure (Figure 11) to avoid confusion 

The selection of 24 antidepressants for this study warrants further explanation. The authors rely heavily on a 2013 study that itself drew from outdated references, raising concerns about the study's novelty and the relevance of the chosen antidepressants. The 2013 study, which served as the primary basis for selecting the antidepressants, appears to have cherry-picked compounds based on limited evidence and without considering more recent advancements in antidepressant research. Given the significant advancements in antidepressant development over the past decade, it is surprising that the authors chose to exclusively rely on compounds identified in a 2013 study.

Answer 1- We would like to thanks the reviewer for valuable comment. We selected 24 anti-depressants based on their mode of action as well as commonly used in the global market. We added one column in the table related to mode of action in Table 1.

Moreover, the authors should conduct additional literature reviews to identify newer and more relevant evidence supporting their choice of antidepressants. This could involve examining recent studies that have investigated the potential of antidepressants in treating Alzheimer's disease or exploring the mechanisms by which these compounds interact with MARK4.

Answer 2 - Thanks for the valuable comment. Based on the comment we added the sentence “Based on the recent meta-analysis finding, sertraline and mirtazapine can be considered as an alternative treatment for depression in AD [18]. Even the combination of antipsychotics (such as risperidone and quetiapine) and mirtazapine can be used for the management of AD [19,20]”. 

The authors' reliance on ADMET analysis to narrow down the pool of potential antidepressants is also a significant concern. While ADMET studies provide valuable insights into the pharmacokinetic and toxicological properties of compounds, they should not be used solely as exclusionary criteria to filter out potential drug candidates.

Answer 3 - Thanks for valuable comment and we apologize for the confusion. We filter out the drugs based on the binding affinities not on ADMET studies. We plotted the radar plot only for top 6 drugs which we got from molecular docking and donepezil. Based on the comment we even added a ADMET analysis table as supplementary 2 and modified the sentence in methodology “As 24 SSRI drugs are already in the market, but still we performed ADMET analysis with the SWISS ADME server which is shown in supplementary 2 and we even plotted the radar graph for the the binding affinity of the top six drugs and donepezil. Even we made changes in result section “In this study, we generated radar plots for the binding affinity of the top six drugs (CID ID - 4184, 2771, 4205, 5533, 4543, and 2160) lies between -9 to -8.2 kcal/mol as well as radar plot of Donepezil which is already approved for AD”.

The authors' reliance on molecular docking to identify promising antidepressants warrants further scrutiny. While molecular docking provides a valuable starting point for identifying potential drug interactions, it is essential to carefully validate the results to ensure their robustness and reproducibility. The authors should compare the predicted binding modes from docking with those obtained from X-ray crystallography as reported in PDB. This will help to establish the accuracy of the docking simulations and ensure that the proposed binding interactions are biologically plausible.

Answer 4 - Based on the comment we added “The conformation of docked Sertraline, Fluoxetine, Escitalopram, Fluvoxamine, Paroxetine and Citalopram with the MARK4 is shown in the Figure 2. The suitable docked pose was selected by mimicking the crystal structure of the MARK4 complex with pyrazolopyrimidine inhibitor (PDB: 5ES1). The docked complex where Sertraline, Fluoxetine, Escitalopram, Fluvoxamine, Paroxetine and Citalopram was present at the same position occupied by pyrazolopyrimidine inhibitor in the crystal structure, was selected for further analysis. Binding free energy of Sertraline, Fluoxetine, Escitalopram, Fluvoxamine, Paroxetine and Citalopram to the MARK4 was found to be - 9 kcal/mol,-8.7kcal/mol, -8.4kcal/mol, -8.3kcal/mol, -8.2 kcal/mol and -8.2 kcal/mol respectively. All six drugs forms several close interactions to active site residues of MARK4 such as Ile62, Lys85, Val116, Met132, Tyr134, Ala135, Leu185, Ala195 and Asp196, and forming one hydrogen bonds with Lys85, and several non-covalent interactions such as Alkyl, Pi-Alkyl and Van der Waals interactions offered by the protein MARK4. It is found that all six drugs are placed at the same position where co-crystal ligand pyrazolopyrimidine inhibitor is placed and it is interacting with the same active site residues, to which co-crystal ligand pyrazolopyrimidine inhibitor is interacting with. The analysis of docked conformations clearly indicates that all six drugs binds deeper into the cavity, and perhaps decrease the accessibility of MARK4 which may be responsible modulation of its biological functions “in the result section of molecular docking.

Justifiy in details the criteria for selecting the top six antidepressants for molecular dynamics simulation and MMGBSA.

Answer 5. Based on the comment, we added “Based on the top six binding free energy of Sertraline, Fluoxetine, Escitalopram, Fluvoxamine, Paroxetine and Citalopram to the MARK4 which was found to be -9 kcal/mol,-8.7kcal/mol, -8.4kcal/mol, -8.3kcal/mol, -8.2 kcal/mol and -8.2 kcal/mol respectively from 24 drugs, we proceed towards further analysis which include molecular dynamics simulations at 100ns followed by MMGBSA” in the result section of molecular dynamics.

Provide a more detailed comparison of the molecular dynamics and MMGBSA results for the six antidepressants.

Answer 6 Based on the comment we modified the MMGBSA result “After analysing the MMGBSA plot, it was observed that for most of the protein-ligand complexes, the values ranged between -40-50 region as shown in orange colour in the plot. Based on the MMGBSA result we can observe that the binding value for both 2771 and 4205 is between -30 -40 which is very similar to the 5SE1-3152 controlled compound whose MMGBSA binding value also ranged between -30-40, respectively. The binding value for 4184, 5533, 4543 and 2160 ranged between -40-50 which is not close to the control drug (donepezil). The MMGBSA result is shown in Figure 11.” We removed the Figure 11 related to MMGBSA to avoid confusion.

6. PLOS authors have the option to publish the peer review history of their article (what does this mean?). If published, this will include your full peer review and any attached files.

---

## [Decision Letter · Decision Letter 1]

8 Mar 2024

PONE-D-23-35218R1Exploring the most promising Anti - Depressant Drug targeting Microtubule Affinity Receptor Kinase 4 involved in Alzheimer’s Disease Through Molecular Docking and Molecular Dynamics SimulationPLOS ONE

Dear Dr. Ahmad,

Thank you for submitting your manuscript to PLOS ONE. After careful consideration, we feel that it has merit but does not fully meet PLOS ONE’s publication criteria as it currently stands. Therefore, we invite you to submit a revised version of the manuscript that addresses the points raised during the review process.

**Most of the comments are addressed. Minor revisions needed to improve the figures quality.**

We look forward to receiving your revised manuscript.

Kind regards,

Sajjad Ahmad

Academic Editor

PLOS ONE

Journal Requirements:

Additional Editor Comments:

Minor revisions needed. Improve figures quality.

Reviewers' comments:

Reviewer's Responses to Questions

**Comments to the Author**

1. If the authors have adequately addressed your comments raised in a previous round of review and you feel that this manuscript is now acceptable for publication, you may indicate that here to bypass the “Comments to the Author” section, enter your conflict of interest statement in the “Confidential to Editor” section, and submit your "Accept" recommendation.

Reviewer #1: All comments have been addressed

2. Is the manuscript technically sound, and do the data support the conclusions?

Reviewer #1: Yes

3. Has the statistical analysis been performed appropriately and rigorously? 

Reviewer #1: N/A

4. Have the authors made all data underlying the findings in their manuscript fully available?

Reviewer #1: No

5. Is the manuscript presented in an intelligible fashion and written in standard English?

Reviewer #1: (No Response)

6. Review Comments to the Author

Reviewer #1: While the author has addressed previous reviewer comments and their work is suitable for publication, the figures require revision due to insufficient resolution, making it difficult to decipher details like legends and color schemes.

7. PLOS authors have the option to publish the peer review history of their article (what does this mean?). If published, this will include your full peer review and any attached files.

Reviewer #1: **Yes: **Shafi Ullah Khan

---

## [Author Response · Author response to Decision Letter 1]

11 Mar 2024

Respected Editor Sir, 

We have uploaded all figure as separate file. Please find the attachments and do the needful.

with regards

Dr. S Rehan Ahmad 

Assistant Professor ,

Department of Zoology ,

Hiralal Mazumdar Memorial College for Women , 

Govt. Aided College , Govt. of West Bengal 

Dakshineswar , Kolkata , West Bengal , India -700035 

Mobile No. +91-8335999175 / +91-9332094070 

E-Mail : zoologist.rehan@gmail.com

Scopus Id : 57226372929

---

## [Editor Report · Decision Letter 2]

13 Mar 2024

Exploring the most promising Anti - Depressant Drug targeting Microtubule Affinity Receptor Kinase 4 involved in Alzheimer’s Disease Through Molecular Docking and Molecular Dynamics Simulation

PONE-D-23-35218R2

Dear Dr. Ahmad,

We’re pleased to inform you that your manuscript has been judged scientifically suitable for publication and will be formally accepted for publication once it meets all outstanding technical requirements.

Kind regards,

Sajjad Ahmad

Academic Editor

PLOS ONE

Additional Editor Comments (optional):

The authors all the concerns and the paper is ready for publication.
---

## [Editor Report · Acceptance letter]

8 Apr 2024

PONE-D-23-35218R2 

PLOS ONE

Dear Dr. Ahmad, 

I'm pleased to inform you that your manuscript has been deemed suitable for publication in PLOS ONE. Congratulations! Your manuscript is now being handed over to our production team.

Kind regards, 

on behalf of

Dr. Sajjad Ahmad 

Academic Editor

PLOS ONE